# Optimal design and operation of Photovoltage distributed generators and shunt compensators for the Vietnam alternative current distribution network to reduce annual energy loss

**Minh Phuc Duong**[1], **Bon Nhan Nguyen**[2*], **Valeriy Arkhincheev**[3], **Thang Trung Nguyen**[1]

**1** Power System Optimization Research Group, Faculty of Electrical and Electronics Engineering, Ton Duc Thang University, Ho Chi Minh City, Vietnam, **2** Faculty of Electrical and Electronics Engineering, Ho Chi Minh City University of Technology and Engineering, Ho Chi Minh City, Vietnam, **3** Buryat Institute of Infocommunications, Siberian State University of Telecommunications and Information Science, Novosibirsk, Russia

\* bonnn@hcmute.edu.vn

## Abstract

This study presents the application of two novel meta-heuristic algorithms, including the Fungal Growth Optimizer (FGO) and Animated Oat Optimization Algorithm (AOO), to optimize the placement of DGs and SCBs for power loss reduction in three alternative current distribution networks (ACDNs), including a practical ACDN with 55 nodes in Vietnam, besides the two IEEE standard ACDNs. In the study, placement of DGs and SCBs on the three grids is conducted in two phases, including the designing phase and the operational phase. In the designing phase, DGs and SCBs were optimized in their placement on the grids by the two algorithms in various scenarios with different DG configurations. The results achieved through scenarios indicate that AOO completely outperforms FGO in terms of the optimal total active power loss values (TAPL) and the stability of the outputs. Particularly, in the two scenarios conducted on the IEEE 69-node, the TAPL achieved by AOO is better than FGO, 3.65% for the first scenario and 72.28% for the second scenario. In the other two scenarios conducted on the IEEE 85-node, AOO still proves its high effectiveness over FGO by reaching the better TAPL at 6.02% in Scenario 1 and 44.88% in Scenario 2. Lastly, AOO is continuously superior to FGO in the four scenarios in the Vietnam ACDN, with 55-node by 1.04% in Scenario 1, 19.52% in Scenario 2, 7.07% in Scenario 3, and 69.60% in Scenario 4. Next in the operational phase, AOO is reapplied to optimize the power output of PVDGs and SCBs within an average day of the first month in the four quarters of the year, with load demand variation and dynamic supplied power from PVDGs in four scenarios with different settings of PVDGs and SCBs. The results clearly reveal that the operational Scenario 4 with PVDGs is configured to supply both active and reactive power to the grid, along with SCBs, which offer the

**Data availability statement:** All relevant data are within the paper and its Supporting information files.

**Funding:** This research was funded by Ho Chi Minh City University of Technology and Engineering, Vietnam, under grant No. T2026-180. The funder played a role in the payment of article publication charge, and report printing.

**Competing interests:** The authors have declared that no competing interests exist.

best value of energy loss in the three quarters, except for the third one, where the operational Scenario 2 is the most suited for energy loss.

## 1. Introduction

### 1.1. Motivation

Alternative current distribution networks (ACDNs) are an important element of a complete power system [1]. ACDNs receive power with low-nominated voltage through the step-down transformers and supply that power to various loads [2]. Nowadays, the unprecedented growth of the economy and population leads to the expansion of ACDNs in both size and capacity to meet the larger power demand [3]. In those circumstances, the total length of ACDNs will be larger due to the increase in the quantity of distribution lines; as a result, the total active power loss (TAPL) across the ACDNs will proportionally increase [4]. In fact, TAPL is one of the crucial factors reflecting the effectiveness of the operation of certain ACDN, besides other factors such as voltage drops, power factors, etc [5]. Reducing power loss is a primary priority in the design, expansion, and planning of ACDNs. An efficient ACDN not only serves as a strong foundation for ensuring the normal operation of loads but also provides significant engineering and economic benefits for the network's managing agent.

### 1.2. Literature review

Reducing power loss in an Advanced Distribution Network (ADCN) can be achieved through three main methods: network reconfiguration [6], installation of Voltage Amplitude Reactive (VAR) compensators [7], and incorporating Distributed Generators (DGs) [8]. Each approach is cost-effective, flexible, and efficient, yet comes with its own pros and cons. Network reconfiguration changes the network's topology to improve performance by rerouting connections through switches [6]. This method effectively reduces power loss, enhances voltage profiles, and balances loads, thereby increasing system reliability [7]. It also assists with fault isolation and restoration during outages, but requires complex calculations and optimization algorithms to determine switch statuses. Modern ADCNs complicate this process due to the integration of various DGs that inject power back into the grid. Placing distributed capacitors, such as Switched Capacitor Banks (SCBs), improves grid efficiency by injecting reactive power [9], thereby enhancing voltage levels and correcting power factors for nearby loads [10]. However, if poorly placed, SCBs can cause over-voltages and unwanted harmonic resonance, leading to equipment damage and reduced efficiency. Therefore, optimizing the size and location of SCBs and DGs is crucial to maximizing their benefits while minimizing risks.

Then, the placement of DGs [11] is another approach that is mostly deployed in modern ACDNs nowadays to achieve different operational objectives, including power loss reduction and voltage profile enhancement. Basically, placing DGs is the deployment of power-generating sources close to loads [12]. This execution allows

less power to be sent through the distribution line, decreasing the burden to the grid in allocating power, reducing the current amplitude in the transmission process, and, as a result, enhancing the overall efficiency, reliability, and security. The authors in [13] used a mixed-integer linear programming (MILP) model to manage reactive power control in medium- to large-scale ACDNs by simultaneously optimizing DG placement and network reconfiguration. Similarly [14,15], combined the placement of DGs, SCBs, and network reconfiguration to minimize power loss and enhance voltage profiles across the grid by employing two meta-heuristic algorithms, including Grey Wolf Optimizer (GWO) and Ant colony algorithm (ACA), to determine the optimal configuration and the best sizes and capacities of DGs and SCBs. By understanding the advantages of meta-heuristic algorithms, the authors in [16] also applied a proposed particle swarm optimization (PPSO) to optimize the placement of SCBs in two standard IEEE ACDN configurations, including 69-node and 85-node, to achieve the optimal value of multiple objective functions, including energy loss reduction, voltage stability, and cost savings. In [17], the combination of DGs allocation and network reconfiguration is conducted in the IEEE 33-node system to reach the minimum value of power loss. Next, the power loss reduction is also the main objective function in [18], in which the authors combined placing SCBs, DGs, and network reconfiguration to achieve the minimum values of the mentioned objective function.

Placing soft open points (SOPs) has been a modern approach that has proven effective in reducing power loss in ACDNs, in addition to the three mentioned earlier. According to [19], SOPs are basically structured by power electronic devices and placed between nodes with no physical connection in ACDNs, offering an exceptional capability to reroute the power flow by connecting or disconnecting those nodes at a certain period to meet the grid operational conditions. However, the SOPs are not a universal approach, meaning that they also have the disadvantages similar to SCBs, DGs, and network reconfiguration such as highly reliant on the controlled power electric devices at real time to ensure the exact operation [20] and high installation cost while deploying the high voltage ACDNs, and suffering a noticeable amount of internal loss [21]. Hence, the deployment SOPs often come along with other approaches in reality, such as DGs [22] or combining with both DGs and SCBs [23] to reach the compromise balance between engineering advances and installation costs. In [24], the placement of SOPs in the large-scale ACDN is combined with different auxiliary devices such as tie switches, on-load tap changer, energy storage system (ESS), and photovoltaic distributed generators (PVDGs) to achieve the minimum value of system loss and mitigate voltage fluctuation during an operational day. Next, the author in [25] executes the optimal location of SOPs using a genetic algorithm (GA) to deal with the negative effects caused by the integration of renewable energy sources (RES) on the IEEE 37-node ACDN, besides their undeniable engineering benefit.

Recently, the presence of PVDGs in ACDNs becomes vital due to the integration of electric vehicle charging stations (EVCSs) to serve the growing demand of transitioning to electric vehicles [26]. The authors in [27] have reduced the power loss and improved voltage stability in the two large-scale ACDNs, including the IEEE 118-node and 415-node, by optimizing the allocation of DGs and SCBs along with network reconfiguration to ensure the system stability with EVCSs. The authors in [28] offer a new framework for optimizing the placement of EVCSs and charging schedules, along with integrating renewable energy sources using different meta-heuristic algorithms. The new framework is then validated in the standard IEEE 33-node ACDN and has proven its effectiveness in real-time consideration. Besides, the proposed PSO has also proven its capability by achieving better power losses and cost reduction compared to others. The intermittency and uncertainties associated with PVDGs also contribute to grid operational instability, which is clearly observed by voltage fluctuation [29]. PVDGs sometimes undersupply power relative to the network's requirement, causing a power shortage and a resulting voltage drop, which directly impacts the operation of connected electrical devices [30]. To address these circumstances, the authors in [31] deployed a Battery Energy Storage System (BESS) on the IEEE 69-node network. This BESS was intended to maximize the benefits offered by PVDGs and Wind-Based Distributed Generators (WBDGs) – in terms of system loss reduction and voltage enhancement – while simultaneously mitigating their aforementioned negative impacts stemming from power fluctuations.

However, the operation of BESSs must be optimized for each period to ensure the proper activation of charge and discharge status in dealing with the uncertainties from PVDGs and WBDGs to maximize their positive effect in power loss reduction and voltage improvement [32]. The authors in [33] offer an effective method for optimizing network reconfiguration and BESS's operation for power loss reduction. BESS is still a relatively new technology that requires complex control in order to achieve the desired precision at certain periods to support the grid operational status [34]. Besides, the investment and operational costs are still high, which might blur the brightness offered by these devices [35]. Furthermore, as noted in previous research, the deployment of BESSs is frequently combined with other devices, such as DGs powered by RESs and SCBs. The consequence of this comprehensive integration is often a longer payback period for the entire RES project, which is detailed in [36]. In fact, the presence of BESS in ACDN configuration provides a significant advantage in balancing the fluctuation of power supplied by renewable energy sources and load demand variation for each period of a whole operational schedule, as indicated in [37]. Furthermore, integrating BESS in the ACDN along with renewable energy sources is also a viable solution in enhancing the grid resiliency and stability and preventing power flow reversal [38].

### 1.3. The main work in this research

Based on the preceding analysis, power loss reduction remains a critical objective for ensuring the planning, expansion, and operational stability of ACDNs. Given the unprecedented growth in load demand driven by economic development, improving ACDN capacity is essential. Instead of relying on the transmission network with higher losses and risks, the preferred strategy involves deploying solutions like SCBs, various types of DGs, and network reconfiguration due to their low cost, flexibility, and rapid deployment. However, deploying these methods in isolation is often suboptimal. Conversely, combining them offers significantly enhanced efficiency, providing superior support for grid stability and reliability while maximizing the primary operational objective. Building upon this understanding, this research provides an academic demonstration concerning the combined optimization of DGs powered by Renewable Energy Sources (RESs) and SCBs. The primary goal is to minimize power loss and enhance voltage profiles. Specifically, the research will optimize the placement of DGs and SCBs across three distinct ACDNs: two standard test systems, including the IEEE 69, 85-node, and one practical ACDN in Vietnam with 55 nodes in various case studies. To confirm the effectiveness of the optimized placement, the 24-hour power loss will be calculated for an average day of each month across the four quarters of a year. The optimization itself will be performed using two novel meta-heuristic algorithms: the Fungal Growth Optimizer (FGO) [39] and the Animated Oat Algorithm (AOO) [40].

The novelties and the main contributions of the whole study can be emphasized by the following terms:

• Successfully apply two novel meta-heuristic algorithms to optimize the placement of PVDGs and SCBs to different ACDNs configurations, including the IEEE 69, 85-node, and a practical one in Vietnam with 55 nodes, for power loss reductions with constant load demand, considering different configurations of DGs

• Present a detailed evaluation and comparison of the effectiveness of the applied algorithms in addressing the considered problem across various criteria. Based on the results and achievements, AOO demonstrates itself to be a powerful and robust search algorithm compared to FGO and the previous ones.

• Presents various strategies for placing PVDGs and SCBs to reduce TAPL, with a detailed evaluation of their practical effectiveness in supporting the design process of the Vietnam ACDN, which has 55 nodes.

• For each of the placement strategies of PVDGs and SCBs on the Vietnam practical ACDN as mentioned above, a 24-hour operational simulation is conducted with load demand variation and dynamic supplied power from PVDGs to provide a detailed view of the actual amount of active power loss reduction at e ach hour.

- Presents a seasonal evaluation of energy loss reduction, based on average daily load profiles and PVDG output on an average day of the twelve months. This analysis highlights the influence of seasonal variations on the effectiveness of the proposed solutions.

Besides the introduction, other contents of this study is structured as follows: "Section 2. Problem formulation" presents the mathematical model of the considered problem with objective function and involved constraints, "Section 3. 3. The Applied Methods" briefly introduces the applied methods focusing on their differences compared to previous others, "Section 4. Results and Discussion" provides the results and the detailed analyses of each scenario, lastly, "Section 5. Conclusions" reveals the important conclusions across the whole research.

## 2. Problem description

### 2.1. Objective function

This study focuses on two single major objective functions, including power loss reduction for one peak hour and energy loss reduction for one year of operation. The one-peak-hour loss reduction and one-year energy loss reduction objectives are expressed as follows:

$$Reduce\ P_{loss,\ peak} = \sum_{a=1}^{N_{Ls}} 3.I_{Peak,a}^2 \times R_a$$

(1)

$$Reduce\ A_{loss,year} = N_{days,c}.\sum_{c=1}^{N_{Ms}}\sum_{b=1}^{N_{Hs}}\sum_{a=1}^{N_{Ls}} 3.I_{a,b,c}^2.R_a$$

(2)

where, $P_{loss,\ peak}$ (in kW) is the active power loss of the peak hour in the grid. $A_{loss,year}$ (in kWh) is the total energy loss of the grid in one year. $I_{Peak,a}$ (in Ampere) is the peak current of the *ath* distribution line corresponding to the peak load in a year. $R_a$ (in Ohm) is the *ath* distribution line's resistance. $N_{Ls}$ is the line number of the grid. $I_{a,b,c}$ is the working current of the *ath* distribution line in the *cth* month at the *bth* hour. $N_{days,c}$ is the day number of the *cth* month in a year. $N_{Ms}$ and $N_{Hs}$ are the month and hour numbers in one year and one day, respectively.

Note that Equation (1) is mainly use to serve the designing phase finding the highest power loss value with peak demand power and the optimal placement of DGs, while Equation (2) is applied to determined the energy loss value in operational phase at each hour of a day, then the energy loss of a month, a year will be subsequently determined.

### 2.2. The considered constraints

The study considers the optimal placement of active and reactive power generation sources in distribution grids, including PVDGs and shunt capacitor banks. The constraints of the distribution grids and the added power sources are as follows:

$$P_{grid,b,c} + \sum_{k=1}^{N_{PVs}} P_{PVDG,k,b,c} - \sum_{m=1}^{N_{Nodes}} P_{Load,m,b,c} - P_{Loss,b,c} = 0$$

(3)

$$Q_{grid,b,c} + \sum_{k=1}^{N_{PVs}} Q_{PVDG,k,b,c} + \sum_{n=1}^{N_{SCBs}} Q_{SCB,n,b,c} - \sum_{m=1}^{N_{Nodes}} Q_{Load,m,b,c} - Q_{Loss,b,c} = 0$$

(4)

$$I_{Peak,a},\ I_{a,b,c} \leq I_a^{Max}$$

(5)

$$P_{grid}^{Min} \leq P_{grid,b,c} \leq P_{grid}^{Max}$$

(6)

$$Q_{grid}^{Min} \leq Q_{grid,b,c} \leq Q_{grid}^{Max} \tag{7}$$

$$P_{PVDG,k}^{Min} \leq P_{PVDG,k,b,c} \leq P_{PVDG,k}^{Max} \tag{8}$$

$$Q_{PVDG,k}^{Min} \leq Q_{PVDG,k,b,c} \leq Q_{PVDG,k}^{Max} \tag{9}$$

$$Q_{SCB,n}^{Min} \leq Q_{SCB,n,b,c} \leq Q_{SCB,n}^{Max} \tag{10}$$

$$2 \leq Lo_{PVDG,k} \leq N_{Nodes} \tag{11}$$

$$2 \leq Lo_{SCB,n} \leq N_{Nodes} \tag{12}$$

Equality constraints (3) and (4) show the active and reactive power balance between supplied and consumed parts. The supplied parts consist of the power grid, PVDGs, and SCBs; meanwhile, the consumed parts comprise the power loss on all distribution lines and the demand of all loads. $P_{grid,b,c}$ and $Q_{grid,b,c}$ are active and reactive power supplied by the power grid in the $cth$ month at the $bth$ hour. $P_{PVDG,k,b,c}$ and $Q_{PVDG,k,b,c}$ are the active and reactive power supplied by the $kth$ PVDG in the $cth$ month at the $bth$ hour. $P_{Load,m,b,c}$ and $Q_{Load,m,b,c}$ are the active and reactive power consumed by the load at the $mth$ node in the $cth$ month at the $bth$ hour. $Q_{SCB,n,b,c}$ is the reactive power supplied by the $nth$ shunt capacitor bank in the $cth$ month at the $bth$ hour. $P_{Loss,b,c}$ and $Q_{Loss,b,c}$ are the active and reactive power losses on all distribution lines in the $cth$ month at the $bth$ hour. The power losses are due to the resistance and reactance of the ath line and are obtained by:

$$P_{Loss,b,c} = 3 \sum_{i=1}^{N_{br}} I_{a,b,c}^2 . R_a \tag{13}$$

$$Q_{Loss,b,c} = 3 \sum_{i=1}^{N_{br}} I_{a,b,c}^2 . X_a \tag{14}$$

where, $X_a$ is the reactance of the $ath$ distribution line.

Inequality constraints (5)-(7) show the constraints of the power grid. Constraint (5) shows the power distribution limit of the ath distribution line, which is connecting two nodes of the grid. The current through the $ath$ distribution line in the $cth$ month at the $bth$ hour ($I_{a,b,c}$) and the peak current ($I_{Peak,a}$) must not be equal to or smaller than the maximum current of the conductor on the ath distribution line. Constraints (6)-(7) show the power supply limits of the power grid at the $bth$ hour in the $mth$ month. Basically, the distribution lines are supplied by a transformer at the slack node, and the power supplied by the transformer is called the grid power. $P_{grid}^{Min}$ and $P_{grid}^{Max}$ are the minimum and maximum active power supplied by the power grid, and $Q_{grid}^{Min}$ and $Q_{grid}^{Max}$ are the minimum and maximum reactive power supplied by the power grid. The power supplied by the grid at the $cth$ hour in the $bth$ month ($P_{grid,b,c}$, $Q_{grid,b,c}$) must be within the lowest and highest supply capacity of the transformer. Normally, the lowest limit of a transformer is 0 MW and 0 MVAr, and the highest limit is the rated power of the transformer. However, in the scope of considering the distribution power grid, the highest limit of the transformer is selected to be the sum of all peak loads.

Constraints (8)-(9) show the generation limit of PVDGs at the $bth$ hour in the $cth$ month. The generation of each PVDG ($P_{PVDG,k,b,c}$ and $Q_{PVDG,k,b,c}$) must be within an acceptable range. $P_{PVDG,k}^{Min}$ and $P_{PVDG,k}^{Max}$ are the minimum and maximum active power supplied by the $kth$ PVDG, and $Q_{PVDG,k}^{Min}$ and $Q_{PVDG,k}^{Max}$ are the minimum and maximum reactive power supplied by the $kth$ PVDG. In some cases, the penetration levels of renewable power sources are limited to be lower than 100%. Here, we consider the maximum penetration levels to be 25%, 50%, 75% and 100%. So, $P_{PVDG,k}^{Max}$ is not higher than 25%, 50%, 75% and 100% of all peak loads. On the contrary, the penetration level of the reactive power sources, including PVDGs and SCBs, can be maximum and equal to 100% of reactive power demand of all loads.

 

Constraint (10) shows each shunt capacitor bank's reactive power generation limit at each hour in each month. $Q_{SCB,n,b,c}$ must be within the lowest and highest limits, which are $Q_{SCB,n}^{Min}$ and $Q_{SCB,n}^{Max}$. $Q_{SCB,n}^{Min}$ and $Q_{SCB,n}^{Max}$ are the minimum and maximum reactive power supplied by the $nth$ shunt capacitor bank. The minimum limit, $Q_{SCB,n}^{Min}$, can be selected to be 0 MVAr; meanwhile, $Q_{SCB,n}^{Max}$ is limited so that the total maximum limits of all SCBs and all PVDGs must be smaller than or equal to the full demand of loads.

Constraints (11)-(12) are about the locations of PVDGs and SCBs in the distribution power grids. $Lo_{PVDG,k}$ and $Lo_{SCB,n}$ are the locations where the $kth$ PVDG and the $nth$ SCB will be installed. Number "2" is Node 2 in the grids where the added PVDGs and SCBs can be installed. Node 1 is the location of the transformer, which can supply full power to all loads when PVDGs and SCBs do not work. $N_{Nodes}$ is the final node in the grid. Here, each PVDG and each SCB can be placed at the same node; however, two PVDGs or two SCBs are not allowed to be placed at the same node.

## 3. The applied methods

This section introduces two algorithms: the Fungal Growth Optimizer (FGO) [39] and the Animated Oat Optimization Algorithm (AOO) [40]. Both are meta-heuristic algorithms that follow similar steps in the optimization process, such as initialization, solution evaluation, and solution refinement. The key differentiator between these algorithms is their method for updating new solutions. We will detail the update procedures for both FGO and AOO in the following subsections.

### 3.1. The fungal growth optimizer

FGO is a novel metaheuristic algorithm inspired by three fungal behaviors: 1) branching strategy, 2) spore germination, and 3) hyphal maturity. The first two behaviors focus on exploration, while the last emphasizes exploitation. Exploration involves searching for potential solutions across the entire search space, whereas exploitation seeks to refine promising solutions in localized areas. FGO has been rigorously tested against four major CEC benchmarks and eleven engineering design problems. Results show that it is competitive or superior to twenty-six state-of-the-art algorithms, establishing FGO as an effective tool for solving complex problems.

FGO executes its update procedure for new solutions using the two phases which are represented the exploration and exploitation capabilities as follows:

- Stage 1: The exploration phase

  FGO executes its exploration phase using the two implementations as given below:

  ◦ The first implementation:

  According to the authors, the newly updated solutions in this first implementation is achieved throughout the relationship between current solution and the others two random ones as described in Eq. (15):

$$S_n^{new\_1} = S_n + e^F \times (S_{sl1} - S_{sl2}) \tag{15}$$

where, $S_n^{new\_1}$ is the $nth$ new solution updated by using the first implementation in the exploration phase, in which $n$ is from 1 to $N_{pop}$ and $N_{pop}$ is the population size that initially set by developer; $S_n$ is the current solution; $S_{sl1}$ and $S_{sl2}$ are randomly selected solutions from the current population $F$ is the gain of the fitness value of the solution $n$ over the whole population and the term $F$ is calculated as follows:

$$F = \frac{Fit_n}{\sum_{n=1}^{N_{pop}} Fit_n} \times rn_3 \times \gamma \tag{16}$$

With

$$\gamma = \left(1 - \frac{I_{Pre}}{I_{Max}}\right)^{\left(1 - \frac{I_{Pre}}{I_{Max}}\right)}$$

(17)

In Eqs. (16)–(17), $rn_3$ is the random values between zero and one; $Fit_n$ is the fitness value of the solution $n$; and $\gamma$ is the shrinking factors; $I_{Pre}$ and $I_{Max}$ are the current iteration and the maximum iteration number.

◦ The second implementation

In this second implementation, the new solutions is updated using two selections with the difference on distance factors denoted by $DF_1$ and $DF_2$ and the growth terms as described in Eq. (18).

$$S_n^{new\_1} = \begin{cases} S_n + \omega_n \times DF_1 + GRF_1, & rn_1 \leq rn_2 \\ S_n + \omega_n \times DF_2 + GRF_1, & otherwise \end{cases}$$

(18)

where $\omega_n$ the amplifying value featured by each solution at certain time of the optimization process. $DF_1$ is the distance factor determined by the distance between the current solution to the best solution; $DF_2$ is another distance factor, determined by calculating the sum of the distance between the current solution to the best solution and the distance between the current solution and a selected solution in the search space; $GRF_1$ is the growth rate which is determined by the using the following expression:

$$GRF_1 = (rn_4 - 0.5) \times rn_5 \times (S_{sl1} - S_{sl2})$$

(19)

where $rn_4$ and $rn_5$ are the random values between zero and one.

• Stage 2: The exploitation phase

Similar to other meta-heuristic algorithms, the exploitation phase of FGO is also highly focused on seeking and refining solutions in local areas, which may be overlooked or neglected during the exploration phase. The exploration phase of FGO is executed using the following mathematical expression:

$$S_n^{new\_2} = \begin{cases} \rho \times S_n + (1 - \rho) \times (S_n + rn_6 \times DF_1 \times GRF_2 + (1 - rn_7) \times GRF_2 \times DF_2), & rn_8 < 0.5 \\ \rho \times S_n + (1 - \rho) \times \left(\frac{\frac{AF+BF}{2} + S_{sl}}{2} + \varepsilon \times rn_9 \times e^F \times \left|\frac{S_{sl1}+S_{sl2}+S_{sl3}}{3} - S_n\right|\right), & otherwise \end{cases}$$

(20)

with

$$GRF_2 = 1 + \frac{Fit_n}{e^{\sum_{n=1}^{N_{pop}} Fit_n}}$$

(21)

$$AF = \frac{I_{Pre}}{I_{Max}} \times S_{GB}$$

(22)

$$BF = \left(1 - \frac{I_{Pre}}{I_{Max}}\right) \times S_{PB}$$

(23)

where, $S_n^{new\_2}$ is the *nth* new solution updated by using the second solution updating method; $\rho$ is the presenting factor, and its value is decided by zero or one deciding the affect of the current solution in creating the process of creating new

solution; $rn_6$, $rn_7$, $rn_8$, and $rn_9$ are the random values between zero and one; $GRF_2$ is another growth factor; $S_{sl3}$ is another random solution in the whole population;s $AF$ and $BF$ are the weigh factors of the global best solution and the local best solution to the process of creating the new solutions; $S_{GB}$ and $S_{PB}$ are the global best across the whole population and the present best at the considered time.

- **Solution method selection condition of FGO**

Generally, the whole update procedure for new solutions of FGO is summarized by the following pseudo code in which the update selection is determined through the four random factors including $rd_1$, $rd_2$, $rn_8$, and $rn_9$. These factors are randomly generated along the process and their values are between zero and one.

If $rn_8 < rn_9$

 The exploration phase will be selected for updating new solutions,

 If $rd_1 < rd_2$

 The frist implementation is applied for updating new solutions,

 Else

 The frist implementation is applied for updating new solutions,

 End.

 Else

 The exploitation phase will be selected for updating new solutions.

End.

## 3.2. The animated oat optimization algorithm

AOO is a novel metaheuristic inspired by the dispersal behaviors of the Animated Oat. It simulates three main mechanisms: (i) seed dispersal by wind, water, and animals; (ii) rolling propagation driven by the seed's awn; and (iii) a propulsion mechanism activated by stored energy when encountering obstacles. AOO operates in two stages: Stage 1 focuses on exploration by modeling natural dispersal behaviors, while Stage 2 emphasizes exploitation by searching around the best solutions. Its effectiveness was validated using the CEC2022 test suite, where it outperformed nine other optimization algorithms, and it was further confirmed across five engineering design problems. The mathematical espression of the two stages are given as follows:

- Stage 1: The exploration phase:

To serve the exploration phase, AOO applies the mathematical model as described in Eq. (24). Based on the equation, the newly updated solution in the exploration phase is dependent on 1) the current solution 2) the additional term and 3) the best solution through the division condition which is determined by the mod of the current index of the solution and the ration of $N_{Pop}/10$ as follows:

$$S_n^{new\_1} = \begin{cases} \dfrac{1}{N_{Pop}} \times \displaystyle\sum_{n=1}^{N_{Pop}} S_n + WF, & mod\left(i, \dfrac{N_{Pop}}{10}\right) = 0, \\ S_{GB}, & mod\left(i, \dfrac{N_{Pop}}{10}\right) = 1, \\ S_n + WF, & otherwise \end{cases} \tag{24}$$

With

$$WF = \frac{\delta}{2} \times (2 \times rn_{10} - 1) \times ub \tag{25}$$

In Eqs. (24)–(25), $WF$ is the additional term determined by Equation (25), and $mod\left(i, \frac{N_{Pop}}{10}\right)$ is the reference terms; $ub$ is the upper boundary of the search space featured by the given optimization problem.

• Stage 2: The exploitation stage

As mentioned earlier, the main purpose of executing the exploitation phase is to take a deeper search in local areas where a high-potential solution for the given problem might exist. AOO executes its exploitation procedure by referencing the current best solution with other terms, such as the integral and the differential terms, as follows:

$$S_n^{new\_2} = \begin{cases} S_{GB} + IG + \delta + LF \times S_{GB}, & rf_1 > 0.5 \\ S_{GB} + DG + \delta + LF \times S_{GB}, & otherwise \end{cases} \tag{26}$$

With

$$IG = (m^2 + L^2) \times \frac{rn_\alpha}{D} \tag{27}$$

$$m = 0.5 \times \frac{rn_{11}}{D} \tag{28}$$

$$DG = \frac{2 \times k \times x^2 \times \sin(2\delta)}{mg} \times \frac{rn_\alpha}{D} \times (1 - \beta) \tag{29}$$

$$k = 0.5 + 0.5 \times rn_{12} \tag{30}$$

$$x = 3 \times \frac{rn_{13}}{D} \tag{31}$$

where $IG$ and $DG$ are the integral and the differential gain determined by Equation (27) and (29), respectively; $\delta$ is the phase angle determined be the trajectory of the solution in the space and its reference $rn_{10}$ is a random value between zero and one $m$, $L$, $k$, and $x$ are the contributing terms; $\beta$ is resitance ratio; $rn_{11}$, $rn_{12}$ and $rn_{13}$ are random numbers, produced between zero and one; $D$ is the dimension numbers featured by the given optimization problem; $rn_\alpha$ is the random value between [-D, D].

• *Solution method selection condition of AOO*

Similar to FGO, the update procedure for new solutions is not a sequential process that utilizes all the equations presented in the exploration and exploitation as described above. There are actually selection mechanisms that are in charge

of determining the procedures that will be selected at a certain point during the whole update process by using the comparison of a selection factor (sf) to 0.5 as described in the following pseudo code:

```
If sf > 0.5

    The exploration phase will be selected for updating new solutions.

Else

    The exploitation phase will be selected for updating new solutions.

End.
```

Note that *sf* in the pseudo code is also another random value between zero and one.

## 4. Results and discussion

In this section, the FGO and AOO are executed to optimize the allocation of DGs and SCBs for power loss reduction on different ACDNs configurations, including the IEEE 69, 85-node [16], and a practical ACDN with 55 nodes in Vietnam. The single-line illustration of the first two ACDNs are displays in Figs 1 and 2, while the practical one will be delivered later in the next subsection. Firstly, the two algorithms are employed to solve the problem from a planning perspective. Then, the better algorithm will be applied to solve the problem from an operational view, using load demand variation and dynamic power supplied from distributed generating sources.

For better proving and comparing the actual performance of the two applied algorithms, both FGO and AOO are set for the same executing parameter in terms of population size ($N_{Pop}$) and maximum iteration index ($I_{Max}$) across all the scenarios as presented in Table 1 below:

Besides the parameters presented in Table 1, FGO and AOO are also executed for 50 test runs for optimal solutions before later comparisons in each scenario. The description for each scenario on each ACDN configuration is given as follows:

i. On the IEEE 69 and 85-node ACDNs

- Scenario 1: Placing three DGs, only injecting active power to the grid.
- Scenario 2: Placing three DGs, injecting both active and reactive power to the grid.

ii. On the Chinfon cement ACDN with 55 nodes:

- Scenario 1: Placing three PVDGs, only injecting active power to the grid.
- Scenario 2: Placing three PVDGs, injecting both active and reactive power to the grid.
- Scenario 3: Placing three PVDGs similar to Scenario 1 along with three SCBs.
- Scenario 4: Placing three PVDGs similar to Scenario 2 along with three SCBs.

＊Note that the results of all the scenarios mentioned in i) and ii) will be delivered in "Section 4.1. The results for the IEEE 69 and 85-node systems", "Section 4.2.Results comparison with previous studies", and Section 4.3. Optimal design PVDGs and switched capacitor banks for Chinfon cement feeder" to serve the designing phase with the main objective function shown in Equation (1) in "Section 2.1. Objective function" and to demonstrate the capability of the applied

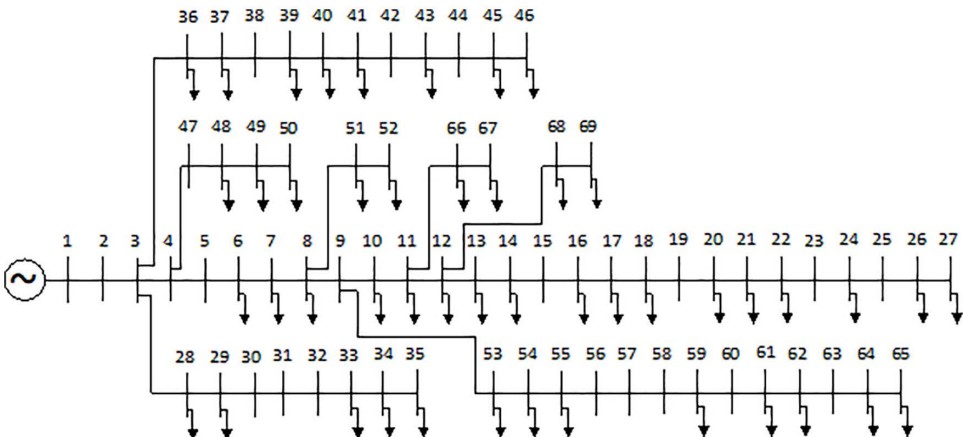

**Fig 1. The illustration of the IEEE 69-node ACDN.**

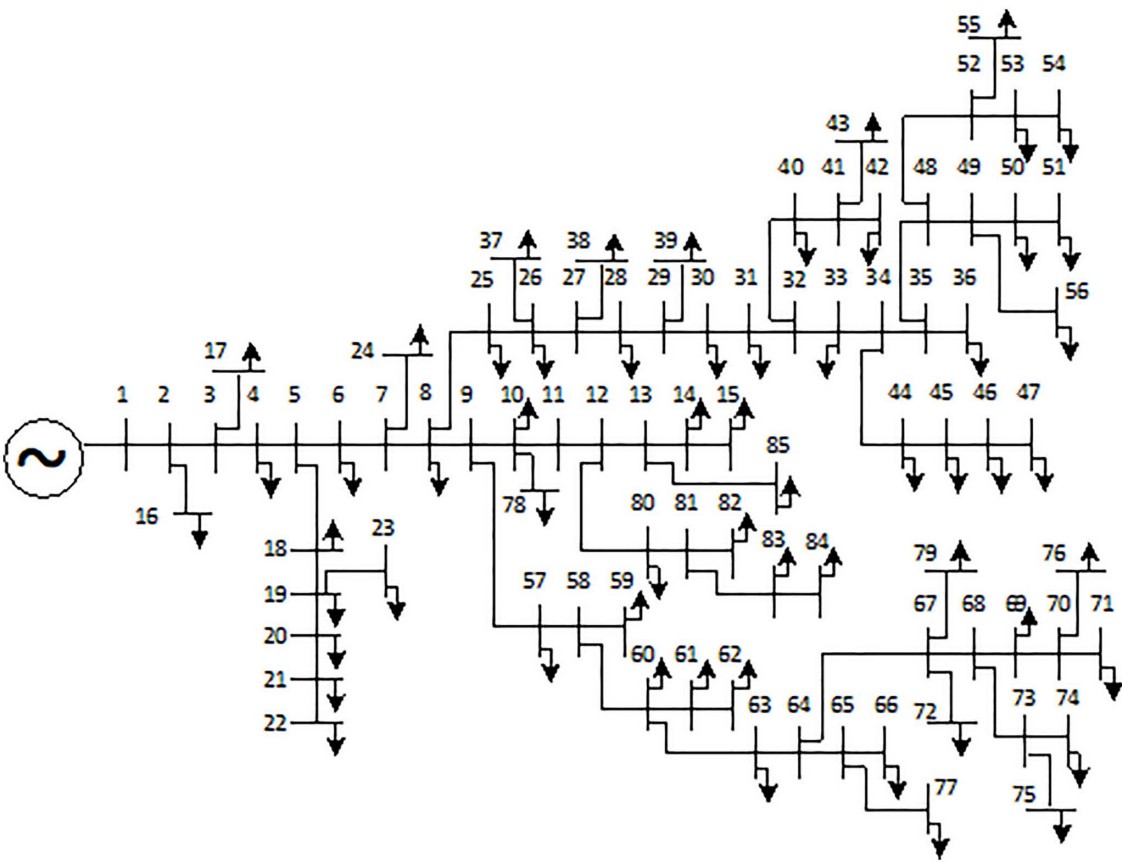

**Fig 2. The illustration of the IEEE 85-node ACDN.**

**Table 1.** The settings of FGO and AOO across all the scenarios conducted within the study.

| Method | | FGO | | AOO | |
|---|---|---|---|---|---|
| **Parmeters** | | $N_{Pop}$ | $I_{Max}$ | $N_{Pop}$ | $I_{Max}$ |
| IEEE 69-node | Scenario 1 | 20 | 200 | 20 | 200 |
| | Scenario 2 | 30 | 300 | 30 | 300 |
| IEEE 85-ndoe | Scenario 1 | 30 | 300 | 30 | 300 |
| | Scenario 2 | 30 | 300 | 30 | 300 |
| Chinfon cement ACDN with 55 nodes | Scenario 1 | 20 | 300 | 20 | 300 |
| | Scenario 2 | 30 | 300 | 30 | 300 |
| | Scenario 3 | 30 | 300 | 30 | 300 |
| | Scenario 4 | 40 | 300 | 40 | 300 |

algorithms, and then only the best applied algorithm will be reapplied in the operational phase in which only the optimal designs achieved in all the scenarios in ii).

This study's work is performed on a personal computer with a 2.26 GHz central processing unit clock speed paired with 16 GB of random-access memory (RAM). Coding and related simulations are also completed using MATLAB programming language version 2018a.

## 4.1. The results for the IEEE 69 and 85-node systems

Fig 3 shows the results achieved by the FGO and AOO for Scenarios 1 and 2. AOO is better than FGO for the two scenario s in terms of the smallest loss and the fluctuation of all fifty runs. Namely, the smallest losses obtained by AOO are 69.426 kW and 4.265 kW for the two scenarios, but those are 72.506 kW and 15.835 kW for FGO. The mean losses obtained by AOO are 70.437 kW and 5.696 kW for the two scenarios, but those are 84.556 kW and 36.062 kW for FGO.

Fig 4 presents the best convergence curves of the FGO and AOO from 50 trial tests, which achieve the optimal value of TAPL. AOO exhibits a significantly faster convergence speed than FGO in both implemented scenarios. FGO, on the other hand, lags behind AOO in achieving similar values. Note that the convergence speed and the ability to reach the optimal solution are two key factors to judge the actual performance of a particular meta-heuristic algorithm. The absence of one of these factors means that a meta-heuristic algorithm cannot be considered a competitive algorithm.

Table 2 offers another aspect regarding the total computing time of the two algorithms while solving the considered problem executed on the IEEE 69-node. The computing times shown in the table are the average time measured in seconds of 50 trial tests. In table AOO, it continuously shows its advantage over FGO in this regard by offering faster computing times in both scenarios implemented in the IEEE 69-node ACDN. In fact, the computing time is also a crucial aspect after the other two, as mentioned above, which are the ability to reach the optimal value of the main objective function and the convergence speed.

Fig 5 presents the results of the two applied algorithms, AOO and FGO, in the second ACDN, the IEEE 85-node network, under two scenarios similar to Fig 3. Despite the larger scale of the problem conducted on the IEEE 85-node compared to the previous IEEE 69-node, AOO still provides unmatched performance compared to FGO in both scenarios. Quantitatively, the superiority of AOO over FGO in the first scenario is 6.02% for the minimum TAPL, 14.93% for the median TAPL, and 27.90% for the maximum TAPL. In Scenario 2, those percentages of AOO over FGO are 44.88%, 60.77%, and 62.59%, respectively.

Fig 6 illustrates the optimal convergence characteristics achieved by FGO and AOO in the two scenarios implemented in the IEEE 85-node ACDN. Despite the large number of nodes compared to the first IEEE 69-node ACDN configuration, AOO still maintains its superiority in convergence speed over FGO. Particularly, AOO only requires around 250 iterations

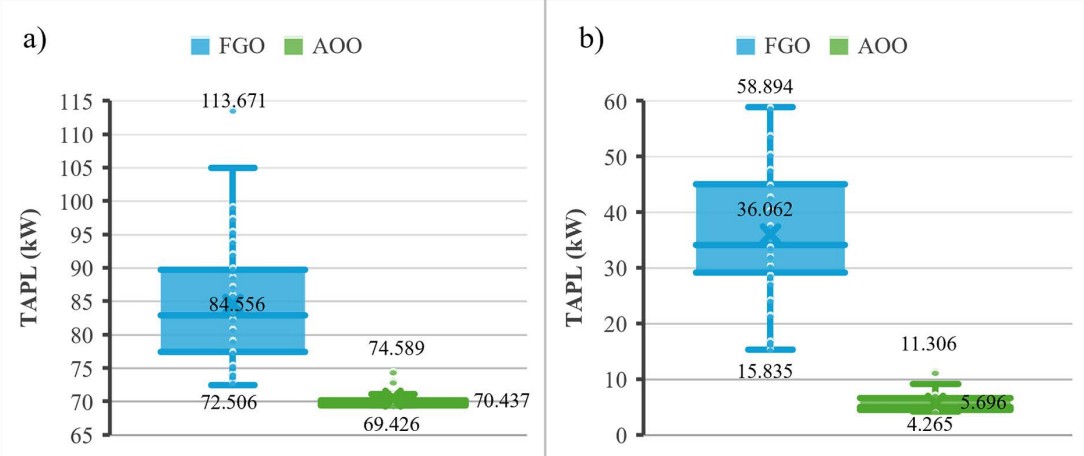

**Fig 3. The results after 50 trial tests for the IEEE 69-node ACDN: a) Scenarios 1 and b) Scenarios 2.**

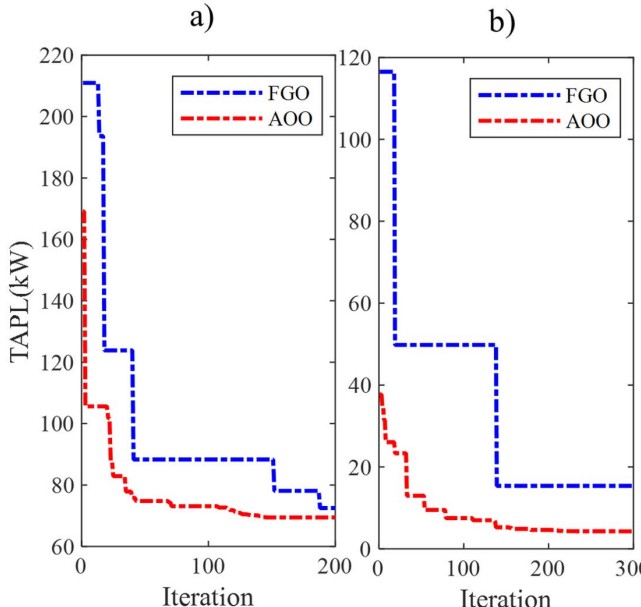

**Fig 4. The best convergence curves of FGO and AOO in a) Scenario 1 and b) scenario 2 in the IEEE 69-node ACDN.**

**Table 2. The computing time (second) of FGO and AOO in two scenarios in IEEE 69-node ACDN.**

| Scenario | FGO | AOO |
|---|---|---|
| 1 | 6.7115 | 6.0025 |
| 2 | 15.4206 | 14.7894 |

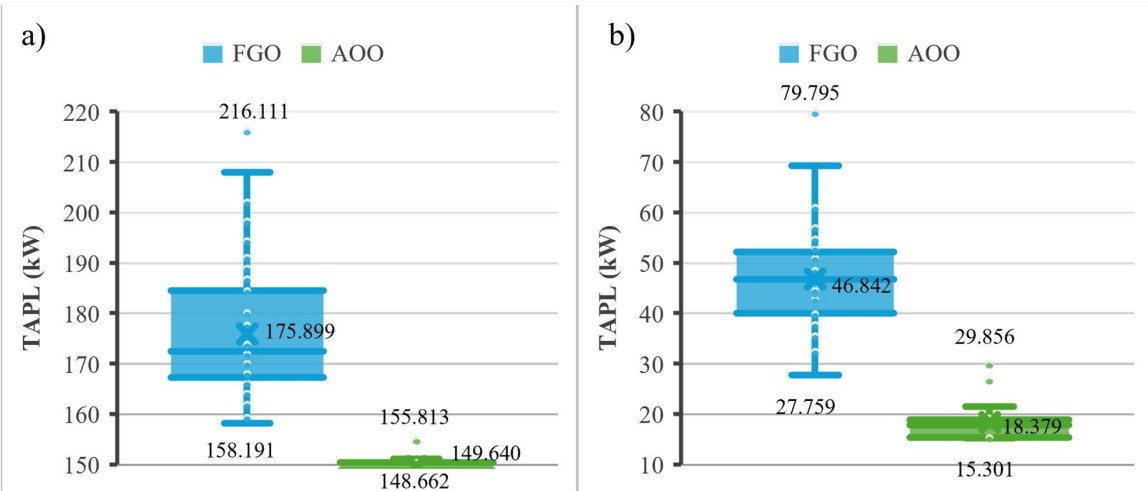

**Fig 5. The results after 50 trial test of the two algorithms in Scenarios 1 and Scenarios 2 on the IEEE 85-node ACDN.**

in Scenario 1 and 230 iterations in Scenario 2 to achieve the optimal TAPL values, while FGO still struggles in both scenarios and remains trapped in local areas, which can be clearly seen in Scenario 2 displayed in Subfigure 6b.

Table 3 presents the average computing times of the two algorithms after 50 trial tests were executed on the IEEE 85-node ACDN. Similar to what is already shown in Table 2 above, AOO still maintains its cutting-edge on the aspect over FGO, regardless of the increase in search space presented by the large quantity of nodes in the IEEE 85-node ACDN compared to the IEEE 69-node ACDN.

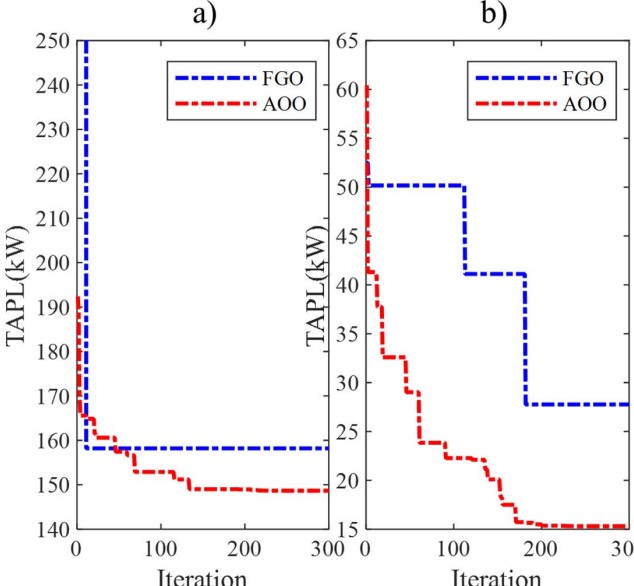

**Fig 6. The best convergence curves of FGO and AOO for the IEEE 85-node ACDN in a) Scenarios 1 and b) Scenarios 2.**

**Table 3. The computing time (second) of FGO and AOO in two scenarios in IEEE 85-node ACDN.**

| Scenario | FGO | AOO |
| --- | --- | --- |
| 1 | 7.070 | 6.6775 |
| 2 | 17.2788 | 16.5800 |

## 4.2. Results comparison with previous studies

This section compares the optimal TAPL values of the FGO and AOO algorithms against other methods from published literature, such as Sine cosine algorithm (SCA) [41], Improved sine cosine algorithm (SCA) [41], Improved symbiotic organisms search algorithm (ISOSA) [42], Moth-Flame Optimization (MFO) [43], Crystal structure algorithm (CrSA) [44], War strategy optimization algorithm (WSA) [44], Average and subtraction-based optimizer (ASBO) [44], Coot optimization algorithm (COOA) [44], Coulomb-Franklin's Algorithm [45], Archimedes Optimization Algorithm (AOA) [45], Transient Search Optimization (TSO) [45], Wild Horse Optimizer Algorithm (WHOA) [45], Modified Firefly Algorithm (MFFA) [46], Coyote Optimization Algorithm (COA) [46], Sunflower Optimization (SFO) [46], Shark optimization algorithm (SOA) [46], Water Cycle Algorithm (WCA) [46], Grasshopper optimization algorithm [47], Improved equilibrium optimizer (IEO) [47]. The comparisons examined the effectiveness of various algorithms in solving the IEEE 69-node and IEEE 85-node ACDN configurations, as illustrated in Figs 7 and 8. The AOO algorithm consistently achieves costs that are equal to or lower than those produced by previously established algorithms. Notably, AOO outperforms SCA [41], ISCA [41], and CRSA [44] in Scenario 1, as well as SCA [41] and CRSA [44] in Scenario 2 for the IEEE 69-node system. In IEEE 85-node system, AOO is more effective than nearly all other algorithms—excluding CFA [45], WHOA [45], and IEO [47]—for Scenario 1,

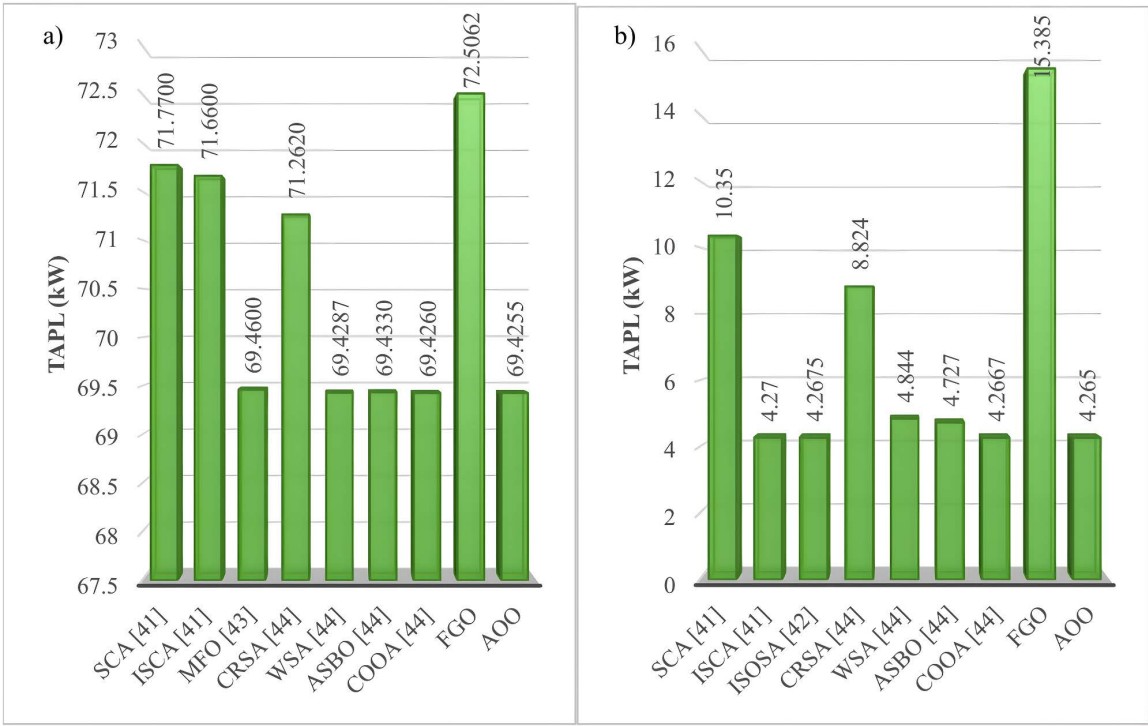

**Fig 7. The result comparisons for the IEEE 69-node system: a) Scenario 1 and b) Scenario 2.**

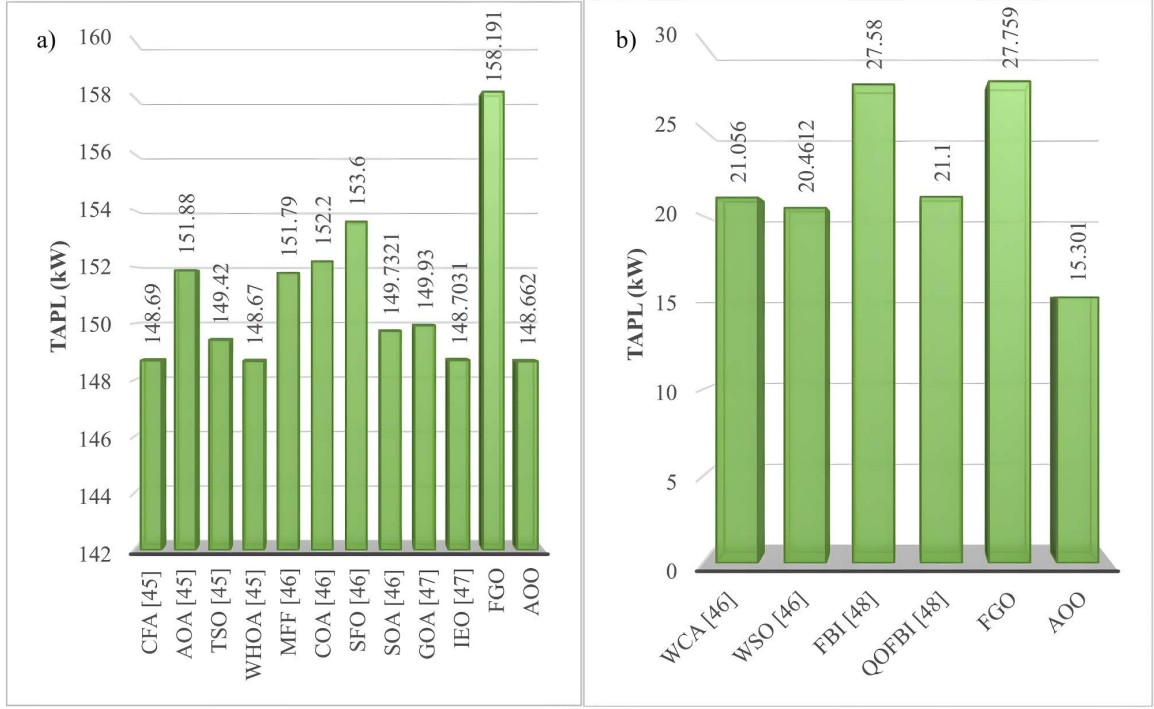

**Fig 8.  The result comparisons for the IEEE 85-node system: a) Scenario 1 and b) Scenario 2.**

and surpasses all algorithms, including Water cycle algorithm (WCA) [46], White shark optimization (WSO) [46], Forensic-based investigation (FBI) [48], and Quasi-oppositional forensic-based investigation (QOFBI) [48] in Scenario 2. The outperformance of AOO compared to other previous methods also comes from its stable performance regardless of the expansion of the space solution, which is determined by the number of desired variables that must be optimized to achieve the best value of the main objective function. Specifically, Table 4 provides an overview of the number of variables that must be determined optimally by AOO in scenarios executed on the two ACDN configurations, including the IEEE 69- and 85-node configurations. The table indicates that Scenario 2 in both ACDN configurations has three more variables needed to determine compared to Scenario 1. Clearly, the overall complexity in Scenario 2 has increased 33.33% over Scenario 1. Furthermore, the increase in the quantity of nodes from 69 to 85 also contributes to the overall complexity of Scenario 2.

### 4.3.  Optimal design PVDGs and switched capacitor banks for Chinfon cement feeder

In this section, FGO and AOO are executed to optimize the allocations of the RDGs along with distributed capacitor banks (SCBs) on a practical ACDN in Vietnam, the 55-node Chinfon cement feeder, which is a part of the whole southern Vietnam ACDN. The single-line diagram of the grid is illustrated in Fig 9. The line and load data of the Chinfon feeder ACDN are reported in S1 and S2 Tables. The type of RDGs chosen to place on the grid is the photovoltaic distributed generators (PVDGs). The PVDGs and SCBs will be optimized on the grid by the two algorithms in four scenarios as follows:

- Scenario 1: Three PVDGs are placed on the grid only, and PVDGs can only supply active power to the grid.

- Scenario 2: Three PVDGs are placed on the grid with the same quantity as Scenario 1, and PVDGs are configured to supply both active and reactive power to the grid.

**Table 4. The quantity of desired values.**

| Variable | The IEEE-69 node ACDN | | The IEEE-85 node ACDN | |
| --- | --- | --- | --- | --- |
| | Scenario 1 | Scenario 2 | Scenario 1 | Scenario 2 |
| $P_{PVDG1}$ (kW) | x | x | x | x |
| $P_{PVDG2}$ (kW) | x | x | x | x |
| $P_{PVDG3}$ (kW) | x | x | x | x |
| $PF_{PVDG1}$ | | x | | x |
| $PF_{PVDG2}$ | | x | | x |
| $PF_{PVDG3}$ | | x | | x |
| $S_{PVDG1}$ (node) | x | x | x | x |
| $S_{PVDG2}$ (node) | x | x | x | x |
| $S_{PVDG3}$ (node) | x | x | x | x |
| Total desired variables | 6 | 9 | 6 | 9 |

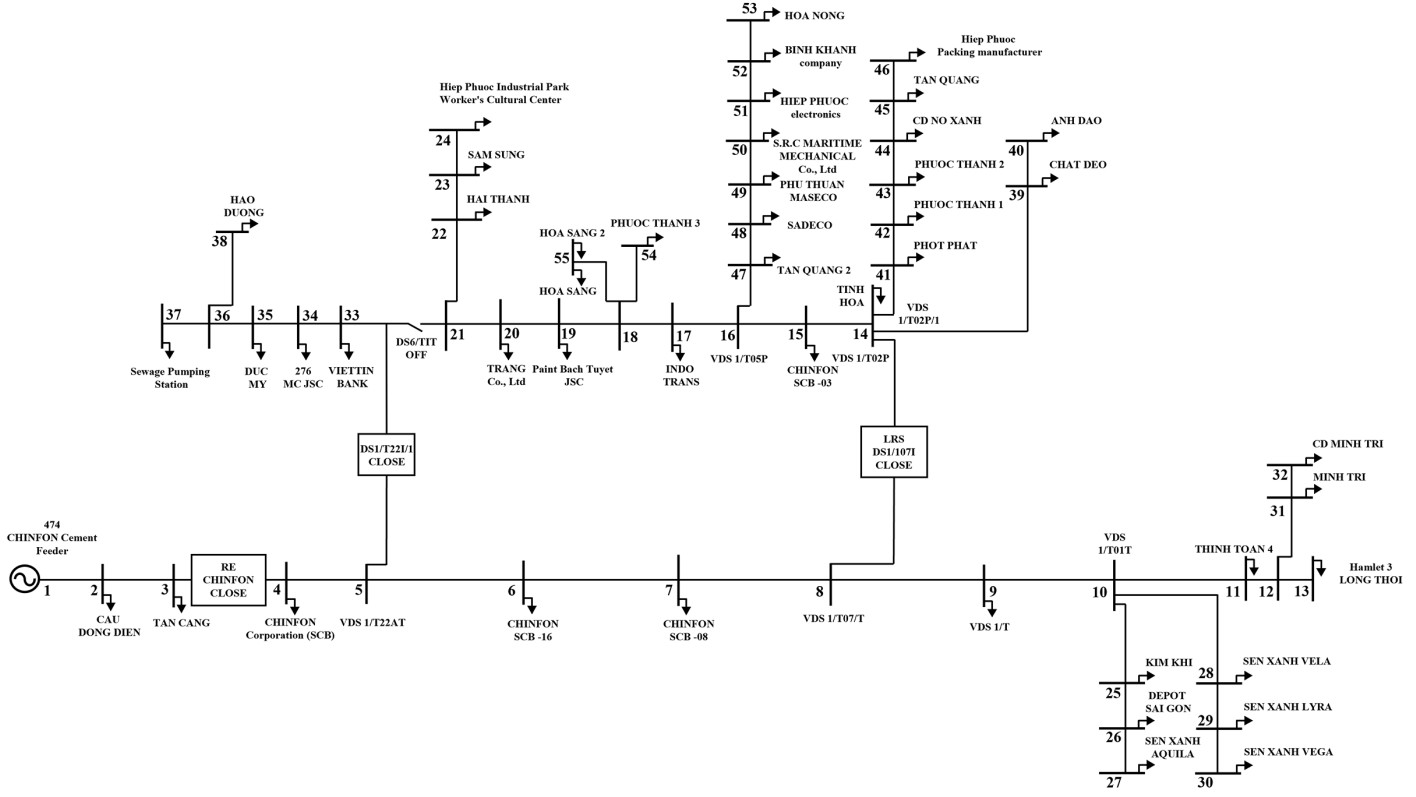

**Fig 9. The illustration of the Chinfon cement feeder ACDN with 55 nodes.**

- Scenario 3: Placing PVDGs with a similar quantity as the first two scenarios, along with three SCBs, and PVDGs can only supply active power to the grid.

- Scenario 4: Placing PVDGs and SCBs as Scenario 3, and PVDGs are configured to supply both active and reactive power to the grid.

The implementation of the four scenarios mentioned above not only provides a reference in designing and planning procedures but also offers a further investigation of the raw performance of the two applied algorithms while tackling the real scenarios, where the solution space gradually increases as described in Table 5.

Figs 10 and 11 clearly show the impressive results. AOO consistently outperforms FGO in all performance metrics, including minimum, median, and maximum TAPL values, while also demonstrating superior stability with narrower box widths and shorter whisker lengths. In the first scenario, AOO achieves minimum, median, and maximum TAPLs of 27.683, 27.755, and 27.932, compared to FGO's 27.974, 28.602, and 29.102. This translates to clear advantages of 1.04%, 2.96%, and 4.02%, respectively. AOO's dominance continues in the second scenario, boasting gains of 19.52%, 46.95%, and 68.28%. In the third scenario, the advantages are 7.07%, 19.03%, and 27.59%. Finally, in the fourth

**Table 5. The quantity of desired variables in the four scenarios.**

| No. | Variable | Scenario 1 | Scenario 2 | Scenario 3 | Scenario 4 |
|---|---|---|---|---|---|
| 1 | $P_{PVDG1}$ | x | x | x | x |
| 2 | $P_{PVDG2}$ | x | x | x | x |
| 3 | $P_{PVDG3}$ | x | x | x | x |
| 4 | $PF_{PVDG1}$ | | x | | x |
| 5 | $PF_{PVDG2}$ | | x | | x |
| 6 | $PF_{PVDG3}$ | | x | | x |
| 7 | $S_{PVDG1}$ | x | x | x | x |
| 8 | $S_{PVDG2}$ | x | x | x | x |
| 9 | $S_{PVDG3}$ | x | x | x | x |
| 10 | $Q_{DCB1}$ | | | x | x |
| 11 | $Q_{DCB2}$ | | | x | x |
| 12 | $Q_{DCB1}$ | | | x | x |
| 13 | $S_{DCB1}$ | | | x | x |
| 14 | $S_{DCB2}$ | | | x | x |
| 15 | $S_{DCB3}$ | | | x | x |
| Total | | 6 | 9 | 12 | 15 |

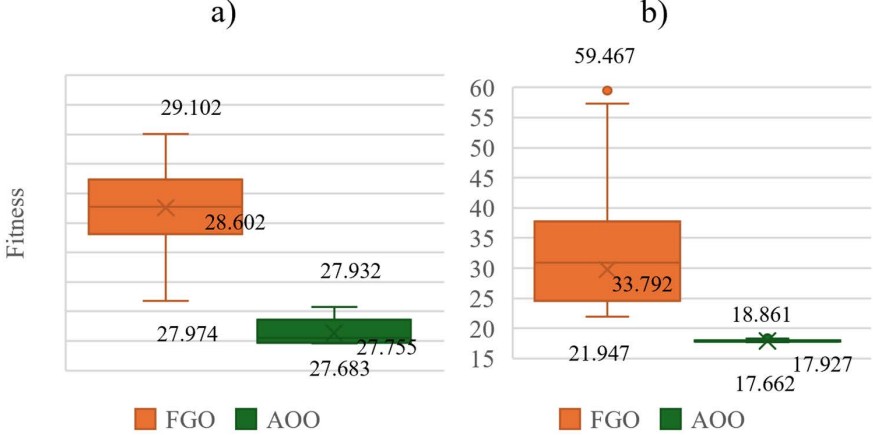

**Fig 10. The TAPL value achieved by the two appplied algorithms in Scenario 1 and Scenario 2 on the Chinfon distribution feeder.**

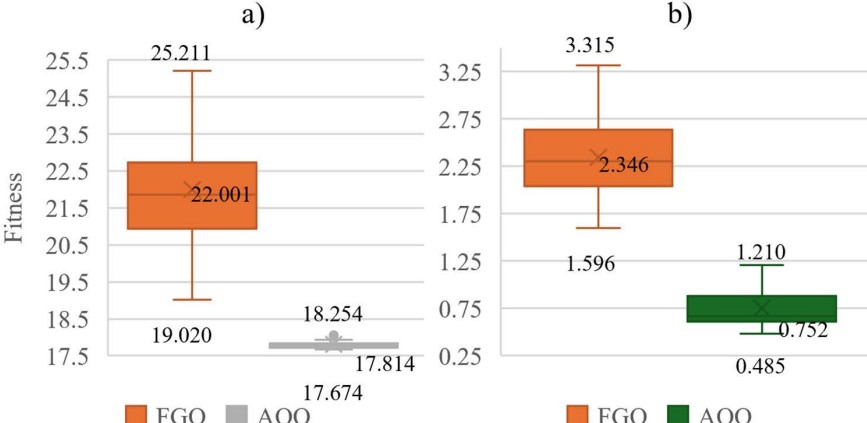

**Fig 11. The TAPL value achieved by the two apppplied algorithms in Scenario 3 and Scenario 4 on the Chinfon distribution feeder.**

scenario, AOO surpasses FGO with gains of 69.60%, 67.93%, and 63.50%. Overall, AOO's narrower boxes and shorter whiskers reinforce its remarkable effectiveness and consistent performance across all scenarios.

Tables 6 and 7 show the optimal solutions and the penetration percentage of PVDGs and SCBs achieved by the AOO in all four scenarios conducted on the Chinfon Feeder ACDN. The observation in Table 7 indicates that Scenario 4 has witnessed a significantly higher active power injected into the grid compared to Scenarios 2 and 3, which are also configured to inject reactive power through the power factor of PVDGs and independent SCBs. In fact, the higher reactive power required in Scenario 4 is the result of prioritizing the power loss values, which is the main objective function, and maintaining the voltage profile as described in "Section 2. Problem description". Particularly, the total active power of Scenario 4 is similar to that in Scenario 1, where SCBs are absent, and PVDGs can only inject reactive power into the grid. The separate integration of SCB and PVDGs has resulted in a significant reduction in power loss values, as shown in Figs 10a and

**Table 6. The optimal solution determined by AOO in the four conducted on the Chinfon Feeder ACDN.**

| No. | Variable | Scenario 1 | Scenario 2 | Scenario 3 | Scenario 4 |
|---|---|---|---|---|---|
| 1 | $P_{PVDG1}$ (kW) | 18 | 20 | 20 | 18 |
| 2 | $P_{PVDG2}$ (kW) | 22 | 22 | 22 | 22 |
| 3 | $P_{PVDG3}$ (kW) | 35 | 54 | 54 | 35 |
| 4 | $PF_{PVDG1}$ | 3000 | 1100 | 1450 | 2900 |
| 5 | $PF_{PVDG2}$ | 2790 | 1850 | 1900 | 2850 |
| 6 | $PF_{PVDG3}$ | 1620 | 700 | 400 | 1650 |
| 7 | $S_{PVDG1}$ (node) | – | 0.7881 | – | 0.75 |
| 8 | $S_{PVDG2}$ (node) | – | 0.7839 | – | 0.856 |
| 9 | $S_{PVDG3}$ (node) | – | 0.806 | – | 0.916 |
| 10 | $Q_{SCB1}$ (kVar) | – | – | 2 | 16 |
| 11 | $Q_{SCB2}$ (kVar) | – | – | 23 | 22 |
| 12 | $Q_{SCB3}$ (kVar) | – | – | 43 | 35 |
| 13 | $S_{SCB1}$ (node) | – | – | 1080 | 2070 |
| 14 | $S_{SCB2}$ (node) | – | – | 1380 | 2250 |
| 15 | $S_{SCB3}$ (node) | – | – | 240 | 1230 |

**Table 7. The penetration level of PVDGs and SCG in the four scenario.**

| Penetration | Active power (PVDGs) | Reactive power (PVDGs) | Reactive power (SCBs) |
|---|---|---|---|
| Scenario 1 | 96.80% | – | – |
| Scenario 2 | 47.68% | 48.73% | – |
| Scenario 3 | 48.99% | – | 46.35% |
| Scenario 4 | 96.67% | 85.86% | 95.28% |

[11b], where the power loss achieved by AOO in Scenarios 1 and 4 is presented in a chart. Next, Scenario 3 also witnesses the separate integration of SCBs along with PVDGs; however, the total reactive power injected into the grid is not as high as that of Scenario 4, as the total amount of active power injected into the grid in Scenario 3 is lower than that in Scenario 4. However, if prioritizing power loss and voltage profile management is maintained, there is engineering logic behind the results in Scenario 3 over Scenario 4. Specifically, Scenario 3 has the presence of both PVDGs and SCBs; however, PVDGs are only configured to supply active power to the grid. Basically, the placement of PVDGs to the grid will results in smaller branch current which directly result in the power loss values, however, the active power supplied by PVDGs must be the optimized values to achieve the best power loss value and satisfies the related constraints, if the PVDGs is set to supply a large amount of active power than the grid actually needed, the extra power will flow to other branches and cause higher current there and then the power loss will inversely increase. Moreover, the extra active power within the grid will affect the voltage in other nodes. To deal with this situation, SCBs are integrated to support the voltage profile and correct the power factor, aiming to maintain the grid's stability. This is clearly evident in Scenario 4, where the grid must adapt to a large amount of active power injected from PVDGs. SCBs are also allocated optimally in proportion to maintain the achievement of minimum power loss and voltage node.

However, the huge injected of active power from PVDGs in Scenario 1 and Scenario 4 might cause the unexpected frequency resonant on the grid due to harmonics. As already seen in Table 7, the amount of active power injected to the grid of those two scenarios is much higher than other remaining ones. Particularly, the injection of reactive power supplied by PVDGs in Scenario 4 requires larger rated power of the inverters that are responsible for providing power factor adjustment by utilizing a combination of power electronic elements linked together in rectifying circuits. Hence, they are considered to be the current harmonic sources that inject extra harmonics into the grid. This additional amount of harmonics, combined with the base amount of the grid, will result in harmonic amplification as a specific frequency contributes to voltage distortion [49]. In an effort to alleviate the unexpected affects from harmonics, there are typical solutions the should be considered while implementing the Scenario 4 in practice as follows:

1) Using the active filter: This method is mentioned in [50] regarding the ability of the inverters that come along with PVDGs to actively detected the unexpected harmonic current order and then generated the similar ones with opposite phase. This method is demonstrated to be effective by reduce the total harmonic distortion (THD) up to 90% while implemented on the DPN with higher presences of renewable energy sources.

2) Apply LCL filter in design phase: This method is mentioned in [51] as a multi-layer filter which consists of an inductor series connected to the inverter, and secondary capacitor and another inductor connected with grid. As proven in the paper, this configuration provides the capability to prevent the undesired frequencies generated by PVDGs in transition periods.

Besides the two solutions above, there are also many others; however, the implementation of any solution must be strictly and carefully considered to achieve the expected efficiency. Otherwise, unexpected damage will lead to the

reduction of the overall stability of the whole system. Particularly, the aspects that need to be considered are the current operational status of the current power, the technical feasibility, and the capital investment.

Fig 12 illustrates the TAPL values achieved by FGO and AOO across four Scenarios. Initially, Fig 12 shows AOO's effectiveness over FGO in all Scenarios, despite differences in PVDG and SCB configurations. Notably, Scenario 4, which uses PVDGs that can inject both active and reactive power, yields the best TAPL value. Scenario 2 and 3 perform similarly, while Scenario 1 shows the most conservative TAPL. It's important to recognize that practical grid operations may require limiting the power supplied by PVDGs and SCBs to maintain stability. Therefore, the optimized configuration from Scenario 4 was tested with varying penetration levels of 25%, 50%, 75%, and 100% relative to load demand. Fig 13 indicates that the 25% penetration level resulted in the highest TAPL due to insufficient compensation, while the 100% level achieved the best TAPL value of 0.631 kW. However, this value is slightly higher than the optimal TAPL obtained without a penetration limit, highlighting the trade-off between loss reduction and operational constraints.

The execution of the four different penetration levels of PVDGs, ranging from 25% to 100%, in the manuscript aims to provide a detailed analysis of adaptivity and engineering boundaries in the operation of the given power grid. Moreover, the penetration level of 25%, 50%, 75%, and 100% provides clear preference points, bolstering the planning process mentioned in [52–55]. According to the previous papers, the 25% penetration of PVDGs provides a baseline measurement of the effect of PVDGs on grid stability and the power loss index compared to other scenarios. In comparison, the 50% and 75% are the prospective preset levels that aim to offer the engineering characteristic variations of the given grid. In practice, integrating 50% and 75% of PVDGs often requires a major upgrade in infrastructure, which also plays a significant role in the later economic considerations. Lastly, the penetration level of 100% is primarily used to assess the engineering adaptability of the given grid in terms of resilience and stability. Furthermore, the penetration of 100% also serves the socio-economic target of achieving zero carbon in most all around the world [55].

Preference to Vietnam policies on planning and integrating the renewable energies to the utility grids, the execution of 25%, 50%, 75%, and 100% of PVDG's penetration is the deduction while considering the circulars No. 39/2015/TT-BCT [56], No. 30/2019/TT-BCT [57] and the National Power Development Plan VIII (PDP8) issued under Decision No. 500/QD-TTg [58], particularly, in the first to Cicurlar regulated that the voltage deviation in the whole grid must be located within +5% to −5% of the rated value. Hence, the four penetration levels from 25% to 100% will support the author in investigating the voltage variations and determining the point where voltage boundaries are violated. Note that voltage is one of the most significant factors that directly affect the operation of devices connected to the grid. Furthermore, the simulated penetrations also allow the authors to have a clear observation and analysis of the overall stability used by the two mentioned circuits. More importantly, the last four penetration levels are also intended to provide a feasible scenario

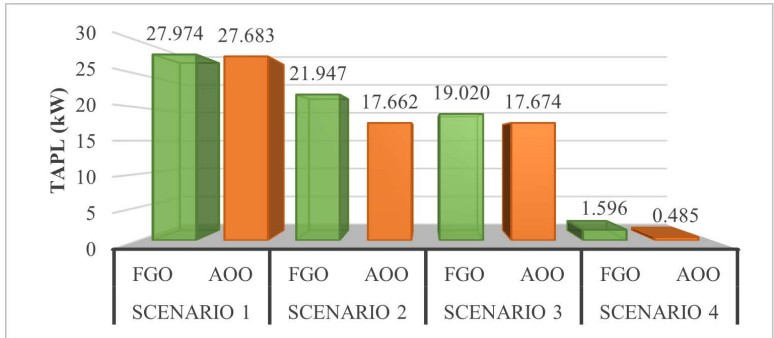

**Fig 12. The TAPL values achieved by FGO and AOO in the four scenario.**

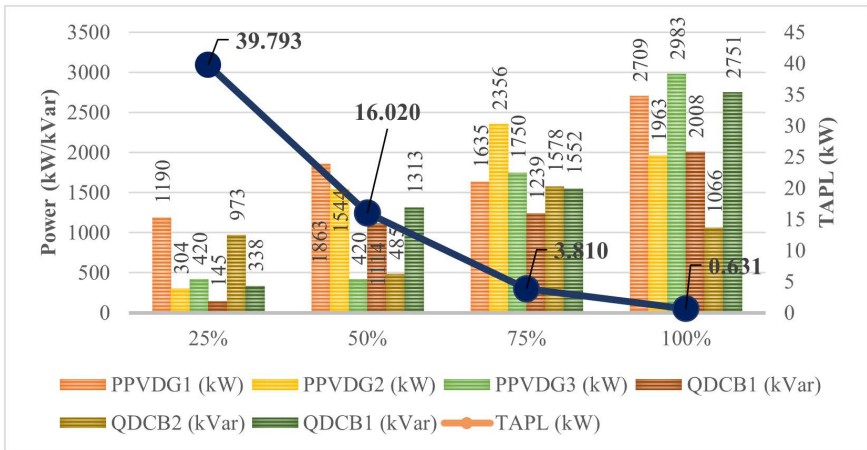

**Fig 13. The amount of active and reactive power injected to the grid in four penetration levels.**

for achieving the long-term target with zero emissions mentioned in Decision No. 500/QD-TTg. Note that the target of zero emission cannot be achieved by immediately integrating 100% renewable energy into the current grid; there must be a careful planning process that gradually increases the penetration of renewable energy to the grid to ensure the balanced and sustainable development of other factors, such as socio-economic conditions, bank loan access, and the improvement of technical and control factors.

Fig 14 illustrates the voltage values obtained at the four penetration levels, with power loss as the primary objective function. The observation from the figure clearly indicates that the minimum voltage nodes achieved on the four penetrations are almost identical, close to 1 p.u., which is the desired value in both planning and operation. These achieved results are largely contributed by AOO, which plays a crucial role in determining the optimal placement of PVDGs and SCBs, ensuring that the voltage value still satisfies the voltage constraints specified in "Section 2. Problem description" and adheres to the specifications issued by Vietnamese regulators. However, the increase in the amount of PV throughout

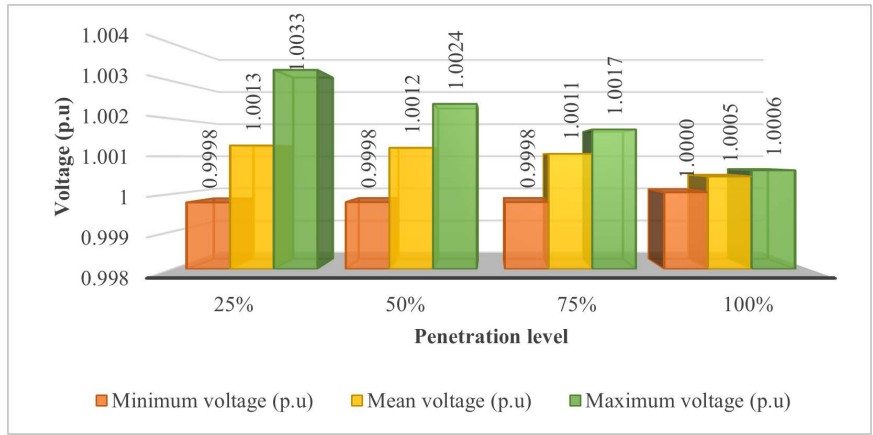

**Fig 14. The voltage summary achieved on the four penetration levels.**

the four penetration levels resulted in a slight decrease in the mean voltage values, and this decrease is more pronounced when observing the maximum voltage nodes.

## 4.4. Optimal operation of the Chinfon cement feeder for different scenarios

In this section, we will reapply the AOO method to optimize the operation of the PVDGs and SCBs designed earlier. Four operational scenarios will correspond to the previously identified scenarios. The power supplied by the PVDGs will vary based on data from the Global Solar Atlas (GSA) at a specific node, with exact values provided for each scenario. Additionally, the reactive power from the SCBs will be adjusted based on the monthly load factors from [59].

### 4.4.1. The results achieved in Scenario 1.
In this section, all PVDGs supply only active power to the grid, as described in scenario 1 of the previous section. Fig 15 illustrates the active power output from the three PVDGs based on real radiation data from GSA [60–62]. Fig 15a shows the output from the PVDG at node 18, while Fig 15b and 15c represent the outputs from nodes 22 and 35, respectively. These outputs reflect the average power on a typical day for each month of the year. Fig 16 displays the power loss over 24 hours for an average day of the twelve months. The performance of PVDGs, shown in Fig 16a, reveals a marked reduction in power loss compared to the scenario without PVDGs in Fig 16b, especially between hours 7 and 18. Fig 17 further details the reduction in power loss observed in Fig 16. Fig 18 indicates that all nodes in the four months have the voltage range from 0.98 to 1.0 Pu; meanwhile, the rated voltage is 1.0 Pu.

The voltage profile achieved in Scenario 1 on an average day of the twelve months is illustrated in Fig 18. In general, all the voltage nodes satisfy the voltage constraint as specified in "Section 2. Problem description", which can only vary within 0.95 and 1.05 p.u. to maintain the normal operation of the whole system.

### 4.4.2. The results achieved in Scenario 2.
This section configures all three PVDGs to supply active and reactive power to the grid. Like the previous section, all the power outputs of the three PVDGs are taken from the GSA [63–65]

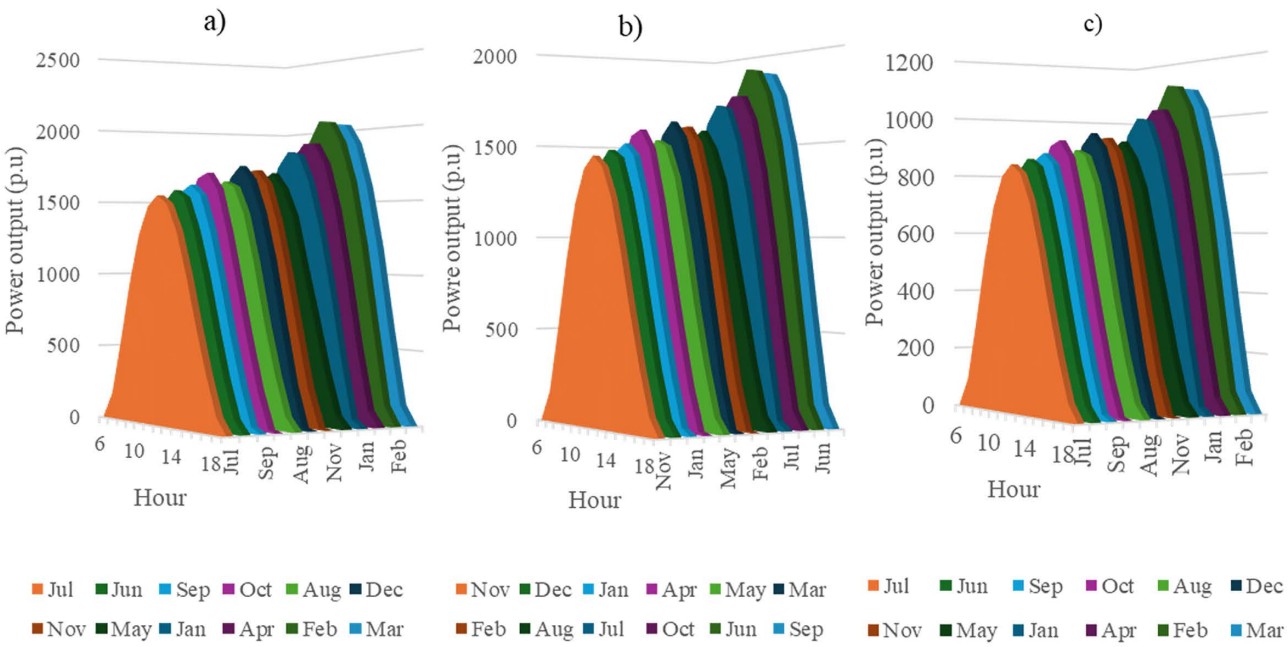

**Fig 15. The power supplied by a) PVDG 1, b) PVDG 2, and 3) PVDG 3 within 24 hours in a average day of each month in Scenario 1.**

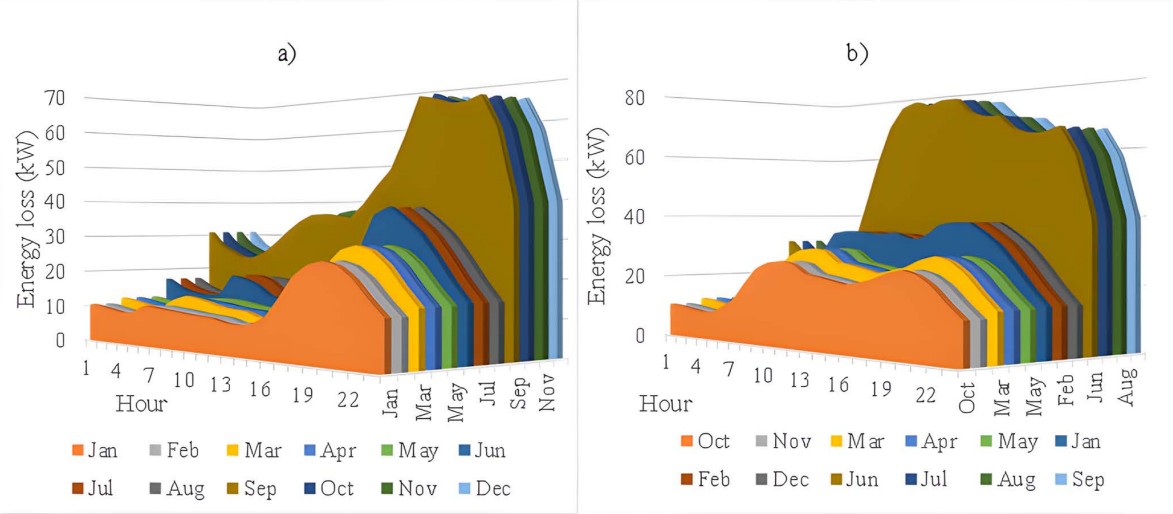

**Fig 16. The power loss value in an average day of the twelve months: a) Scenario 1, and b) the base system.**

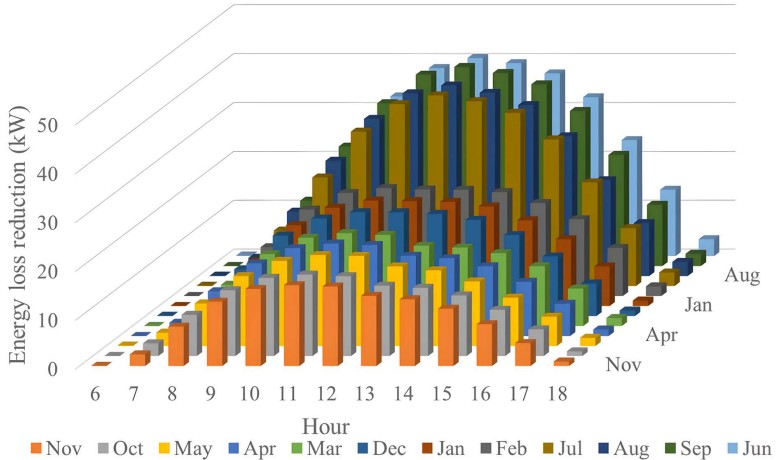

**Fig 17. The power loss reduction at each hour in an average day of the twelve months in the Scenario 1.**

corresponding to the optimized locations found by AOO in Scenario 2 in "Section 4.3. Optimal design PVDGs and switched capacitor banks for Chinfon cement feeder" and presented in Fig 19. Fig 19a, 19b, and 19c, respectively, present the power output of the first, second and third PVDG.

Fig 20a and 20b illustrate the power loss values for each hour of an average day of the twelve months, both with and without the presence of PVDGs. The data indicates that the inclusion of PVDGs results in a significant reduction in power loss during the day when all PVDGs are active. However, it is important to note that power loss values remain unchanged at night, as observed from hours 1–5 and from hours 19–24. Fig 21 provides a closer look at the extent of power loss reduction during the daytime across the four months. Of these months, July experiences the most considerable reduction in power loss, while October shows the least reduction.

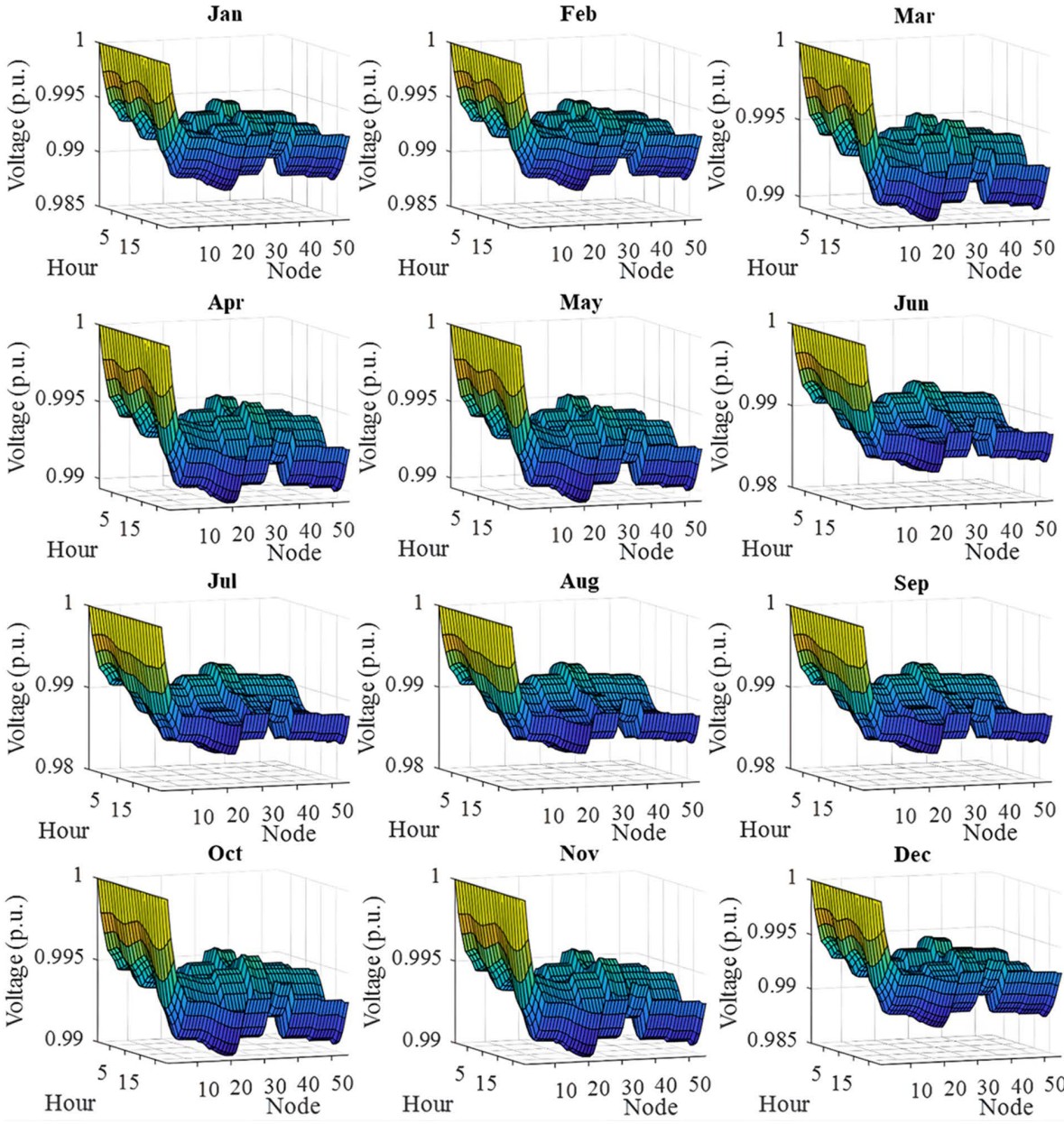

**Fig 18. The voltage profile within 24 hours in an average day of the twelve months in a year achieved in Scenario 1.**

Fig 22 shows the voltage profile from AOO in Scenario 2 during the twelve months within a year. The observation on the figure indicates that all the voltage profile share a common shape and satisfies the voltage contraints specified in "Section 2. Problem description".

**4.4.3. The results achieved in Scenario 3.** In this section, all the PVDGs optimize their operation for 24 hours in an average day of each first month in the four quarters along with SCBs. Unlike the first two scenarios, the PVDGs in this scenario are configured to inject the active power loss to the grid, while SCBs are in charge of supplying reactive power to the grid. The PVDGs in this scenario are placed at nodes 20, 22, and 54, while the SCBs are connected with nodes 2, 23

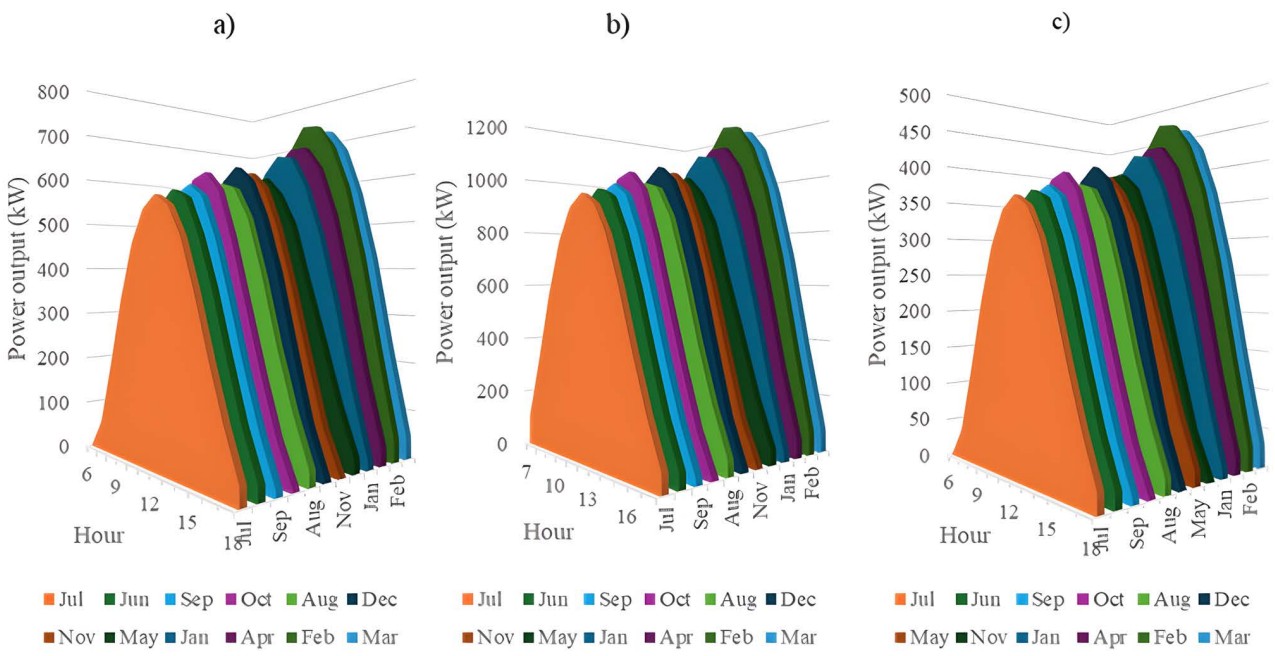

**Fig 19. The power supplied by a) PVDG 1, b) PVDG 2, and 3) PVDG 3 within 24 hours in a average day of each month in Scenario 2.**

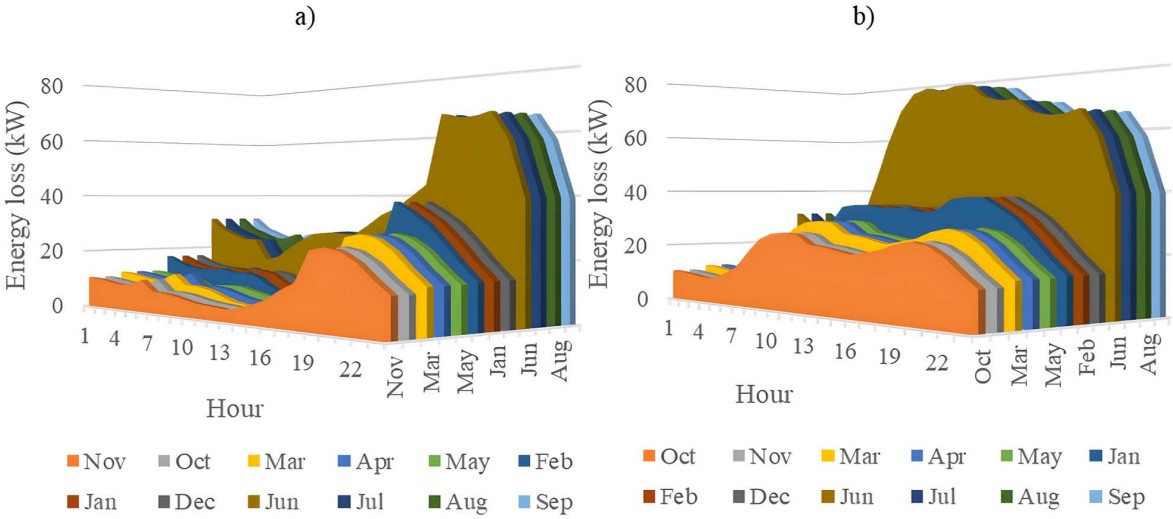

**Fig 20. The power loss value in an average day of the twelve months: a) Scenario 2, and b) the base system.**

and 42. The reactive power supplied by SCBs varies in accordance with the load factors, while the supplied power of the three PVDGs is derived from GSA [66–68] and illustrated in Fig 23.

Fig 24a and 24b show power loss values in Scenario 3, comparing systems with and without PVDGs and SCBs. The inclusion of these components significantly reduces power loss during daytime hours (from hour 7–18), and even during

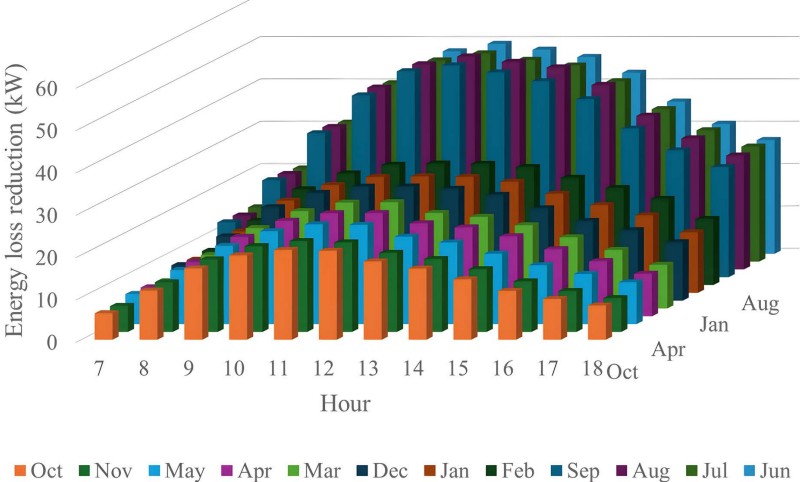

**Fig 21. The power loss reduction of each hour of the four months in the Scenario 2.**

off-peak hours, the voltage support from SCBs helps decrease power loss further by lowering current flow in distribution lines. Fig 25 details the hourly reduction in power loss. While SCBs do contribute to loss reduction at night, their impact is less significant than when combined with PVDGs during the day. This highlights the effectiveness of separating active and reactive power supply. In Scenario 2, unlike Scenario 3, both active and reactive power are injected into the grid. However, reactive power supply is restricted to the daytime when PVDGs are active. At night, with PVDGs offline, reactive power injection is disabled, leading to no voltage support and unchanged power loss. Fig 26 reveals that the system has a good voltage profile from 0.985 to 1.0 Pu.

**4.4.4. The results achieved in Scenario 4.** This section describes the simultaneous connection of PVDGs and SCBs to the grid, similar to Scenario 3. The PVDGs provide both active and reactive power—active power is based on GSA data [69–71], while reactive power depends on their active output and power factor. The SCBs' reactive power varies with daily load factors. The three PVDGs are linked to nodes 17, 22, and 35, with their 24-hour active power output shown in Fig 27.

Fig 28a and 28b illustrate the 24-hour power loss values for an average day in the first month of each quarter, comparing systems with and without PVDGs and SCBs. The data shows that using PVDGs to supply both active and reactive power, along with SCBs, leads to significantly lower hourly power loss than in Scenario 3, where PVDGs only supplied active power. This advantage is particularly notable in July and October. Fig 29 further details the hourly power loss reductions, indicating that the combined capabilities of the PVDGs and SCBs greatly enhance overall effectiveness compared to Scenario 3. Fig 30 reveals that the system has good voltage profile from 0.995 to 1.0 Pu.

Fig 31 show the active power used to achieve the optimal value of the main objective function, compared to the total power supplied by the GSA from three PVDGs in Scenario 4 across the twelve months. The figure emphasizes the optimization algorithm's vital role in selecting appropriate power levels to minimize the main objective while ensuring that grid parameters like voltage and current stay within acceptable limits.

### 4.5. The comparison of the simulation scenarios

This section compares four operational scenarios for January, April, July, and October, as shown in Fig 32. The aim is to identify the most effective approach for managing load variations throughout the year. Scenario 4, which includes both PVDGs and SCBs, is the best for reducing power loss. In this scenario, the PVDGs provide both active and reactive power, unlike Scenario 3, which only generates active power. Interestingly, for July, Scenario 1—utilizing only PVDGs

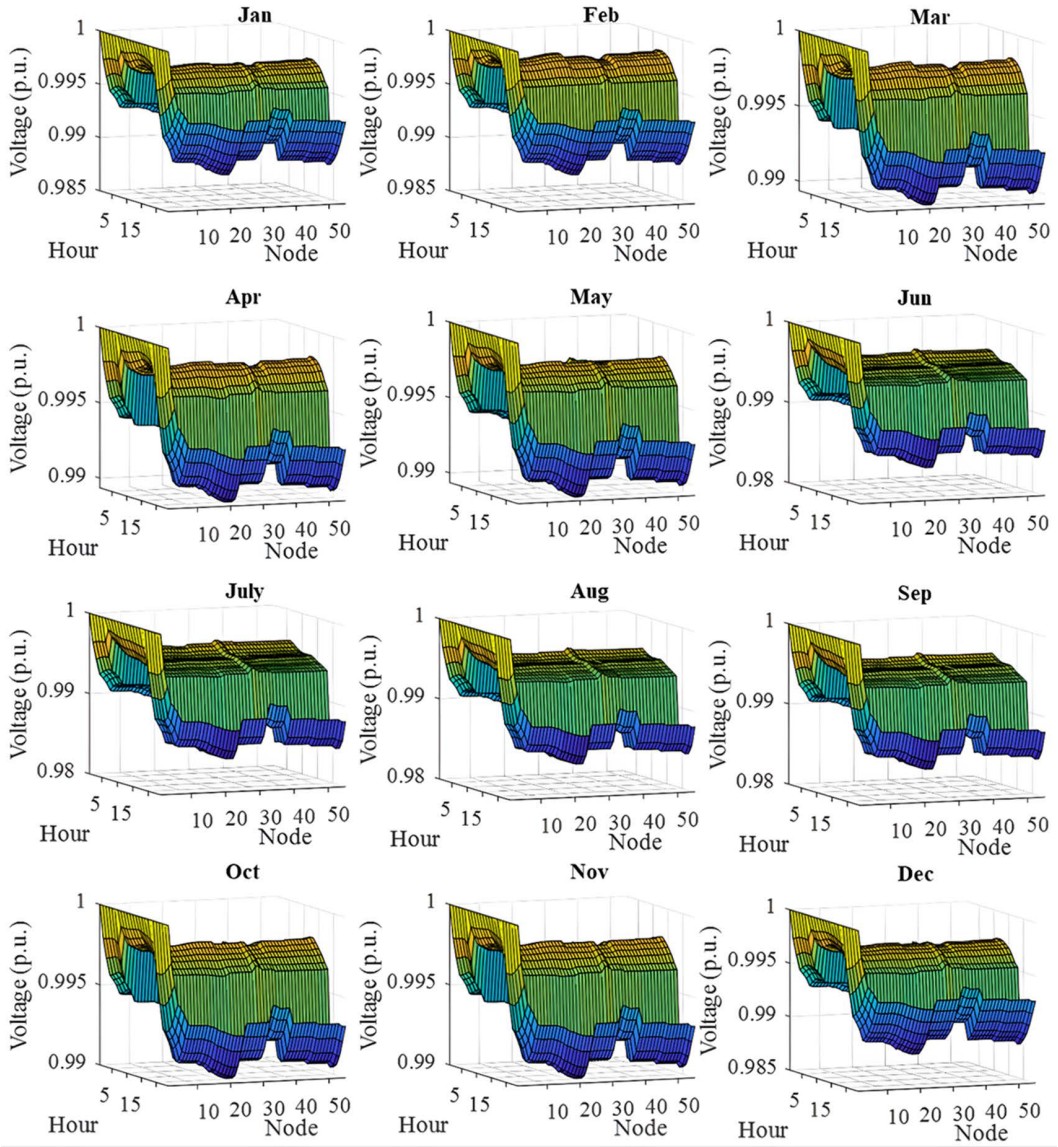

**Fig 22. The voltage profile within 24 hours in an average day of the twelve months in a year achieved in Scenario 2.**

capable of supplying active power only—proves to be the best option for minimizing power loss. The better efficiency of Scenario 1 compared to Scenario 4 in June can be explained by considering the differences in the implementation of these two scenarios. Specifically, Scenario 1 achieved energy loss by contributing active power to the grid through PVDGs only. In contrast, Scenario 4 achieved energy loss by utilizing both active and reactive power from PVDGs and SCBs. In Scenario 1, the injection of only active power from PVDGs will directly result in a proportional reduction in branch current. Scenario 4, on the other hand, has the extra reactive power from SCBs, plus the amount supplied by PVDGs, along with the active power injected into the grid. Moreover, July is the month with the highest load profile, as shown in [59], along with lower power supplied from PVDGs, as displayed in Fig 24. The supply of active power from PVDGS in Scenario 4 still offers the better energy loss in daytime compared to Scenarios 2 and 3; however, Scenario 4 also faces the extra injection of reactive power, which cannot be effectively handled by the grid and contributes to the increase of branch current,

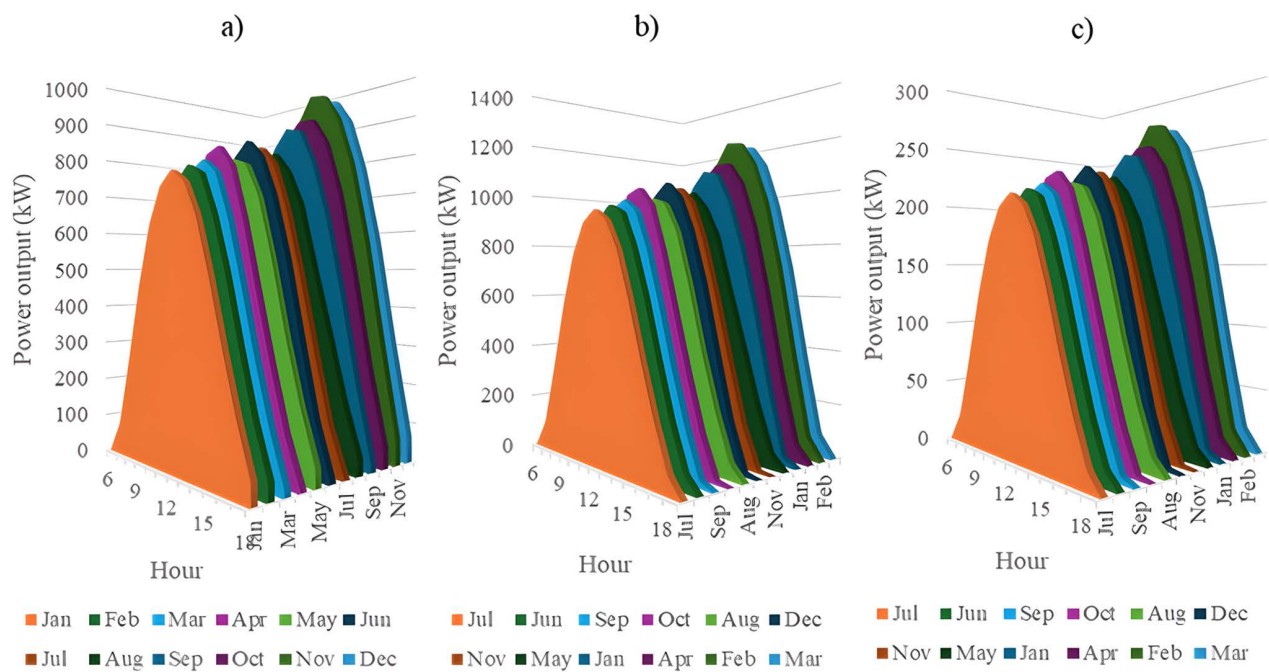

**Fig 23. The power supplied by a) PVDG 1, b) PVDG 2, and 3) PVDG 3 within 24 hours in a average day of each month in Scenario 3.**

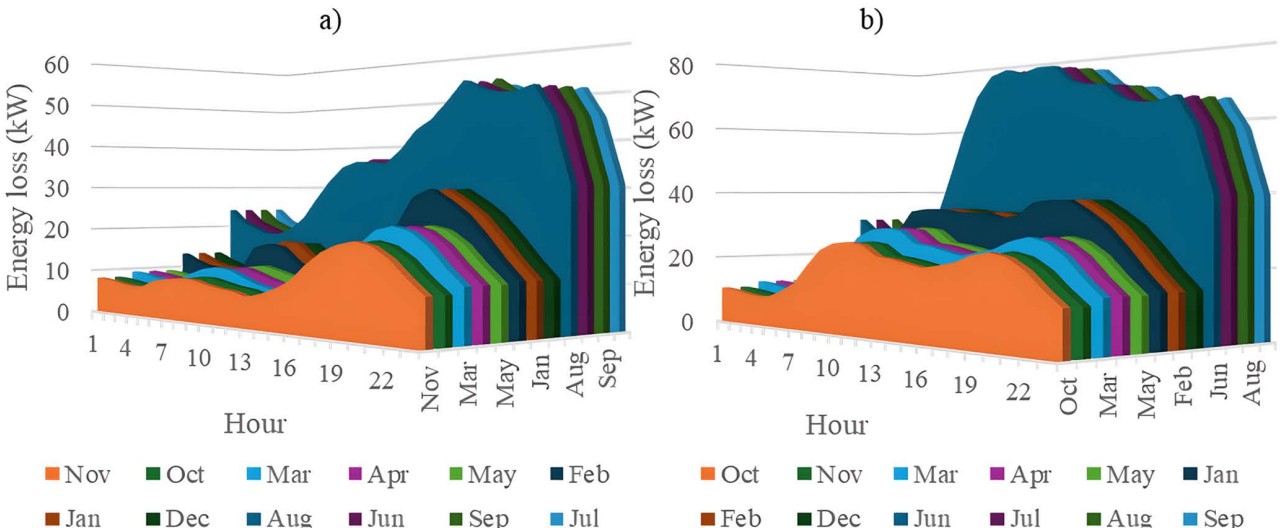

**Fig 24. The power loss value in an average day of the twelve months: a) Scenario 3, and b) the base system.**

resulting in higher energy loss. Preparing various operational scenarios is crucial for effective grid management, allowing for better resource allocation and quicker fault resolution. This helps mitigate economic losses and improve customer satisfaction. However, creating these scenarios requires significant upfront investment. A thorough planning process and cost-benefit analysis are essential to balance investment with system safety and reliability, optimizing spending while avoiding waste.

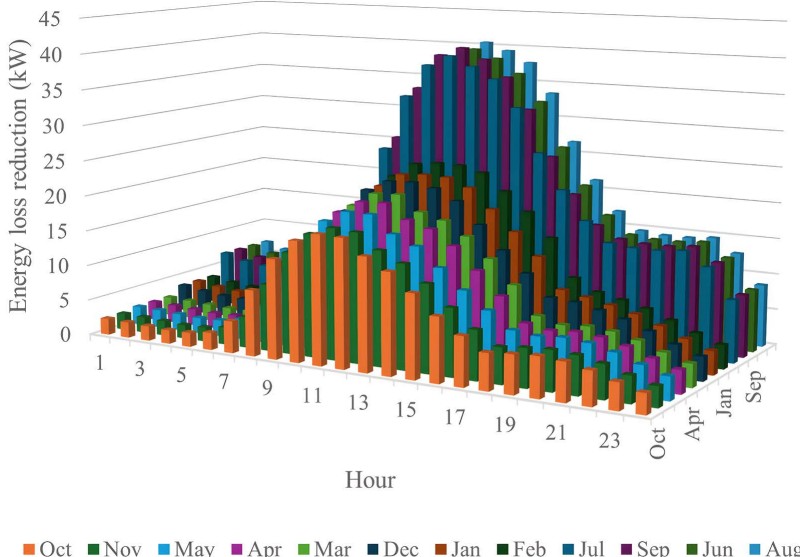

**Fig 25. The power loss reduction of each hour of the four months of the Scenario 3 compared to the base system.**

## 4.6. The discussion on the economic feasibility of the PVDGs project in Scenario 4

In this section, a series of calculations regarding the economic assessment of implementing all the PVDGs with the designed parameters is shown in Table 6 of "Section 4.3. Optimal design PVDGs and switched capacitor banks for Chinfon cement feeder". The calculation aims to provide a detailed look at the total cost of a PVDG with a considered lifetime of 15 years, compared to the original configuration of the grid where the PVDG project is not implemented. Particularly, the total cost of these two scenarios is achieved through the following mathematical models:

$$TC_{Bf} = Y \times \sum_{c=1}^{12} N_{day,c} \sum_{b=1}^{24} \left( P_{Loss,b,c} + \sum_{m=1}^{N_{Nodes}} P_{Load,m,b,c} \right) \times Pr_b \tag{32}$$

$$TC_{Af} = \sum_{y=1}^{15} \left( \sum_{c=1}^{12} N_{day,c} \sum_{b=1}^{24} \left[ \left( P_{Loss,b,c} + \sum_{m=1}^{N_{Nodes}} P_{Load,m,b,c} \right) - P_{PVDGs,b,c} \right] \times Pr_b \right)_y + EPC_y \tag{33}$$

Where $TC_{Bf}$ and $TC_{Af}$ are the total cost before and after integrating PVDGs by USD; $N_{day,c}$ is the number of days of the $cth$ month; $Pr_b$ is the electricity price at $bth$ hour, which is called Time-of-Use Tariff (TOUT) [72] and shown in Table 8; $P_{PVDGs,b,c}$ is total power output in kW of all installed PVDGs at $bth$ hour in the $cth$ month. $EPC_y$ is the entire project cost in $yth$ year with $y = 1, 2, …, Y$ and $Y$ is the considered project's lifetime.

After both $TC_{Bf}$ and $TC_{Af}$ are fully determined, the economic feasibility (*EF*) of the entire projects will be determined as follows:

$$EF = TC_{Bf} - TC_{Af} \tag{34}$$

It is very clear to observe that the main difference between $TC_{Af}$ and $TC_{Bf}$ is in the expression of $P_{PVDGs,b,c}$ and $EPC_y$. While $P_{PVDGs,b,c}$ is the amount of power output from the PVDGs that has already been shown in the previous section,

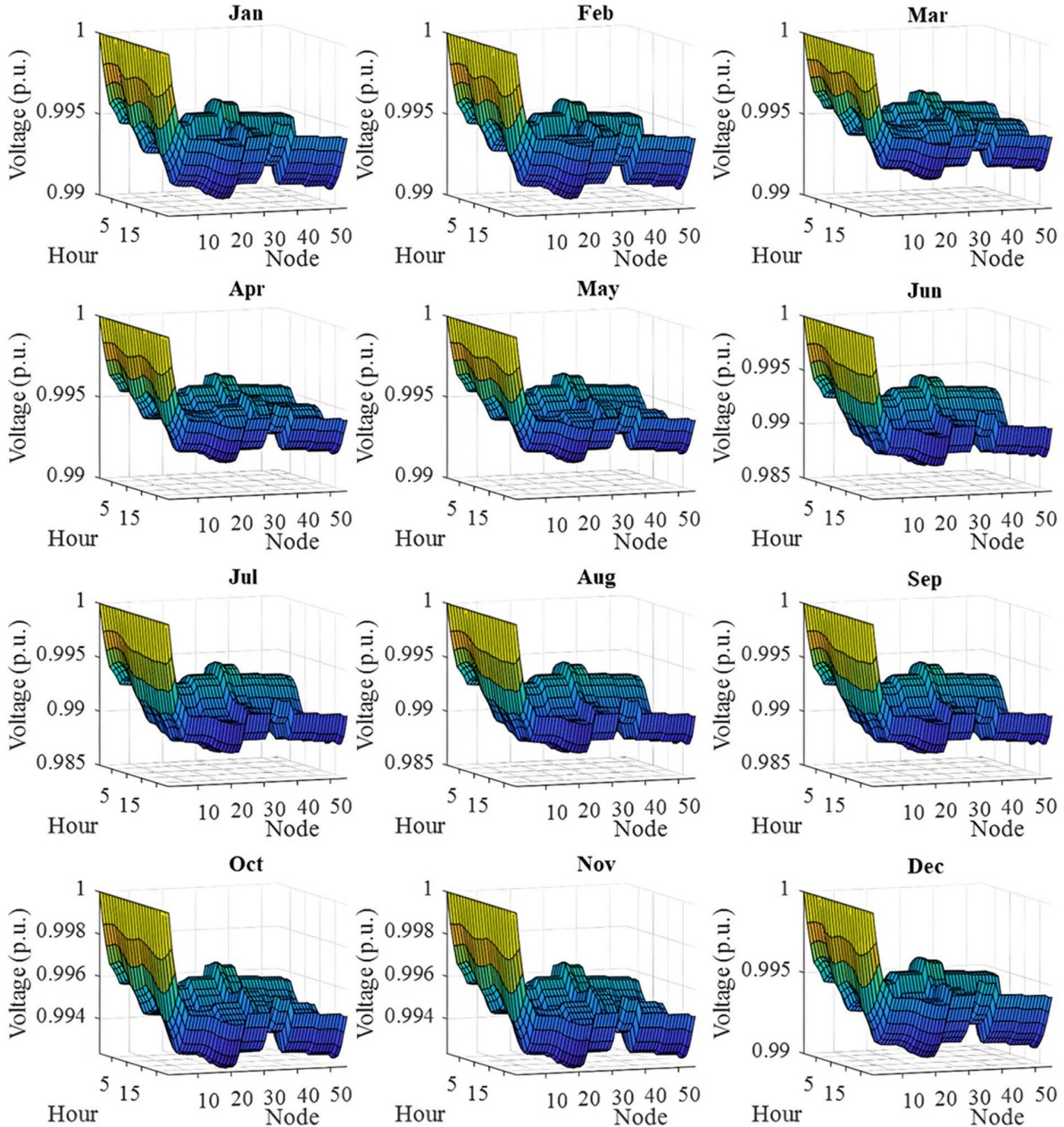

**Fig 26. The voltage profile within 24 hours in an average day of the twelve months in Scenario 3.**

$EPC_y$ is the total annual cost of each year throughout the lifetime of the whole project, and the term is determined by applying the following expression:

$$EPC = CAPEX + \sum_{y=1}^{Y} \frac{Pr_{O\&M}^{Fixed} \times PR_{PVDG}}{(1+r)^y} + \sum_{y=1}^{Y} \frac{Pr_{O\&M}^{Varied} \times Yi_y}{(1+r)^y} + \sum_{InvRep} \frac{C_{InvRep}}{(1+r)^y} + \frac{C_{Degr}}{(1+r)^y}$$

(35)

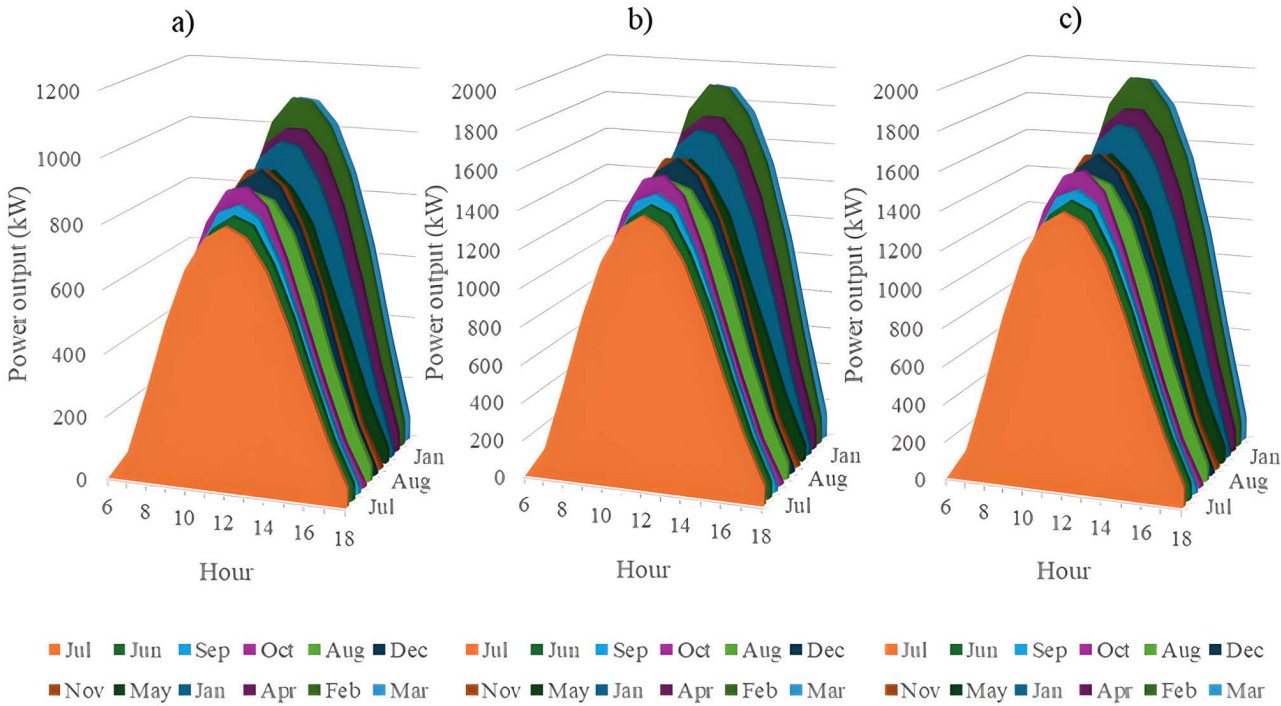

**Fig 27. The power supplied by a) PVDG 1, b) PVDG 2, and 3) PVDG 3 within 24 hours in a average day of each month in Scenario 4.**

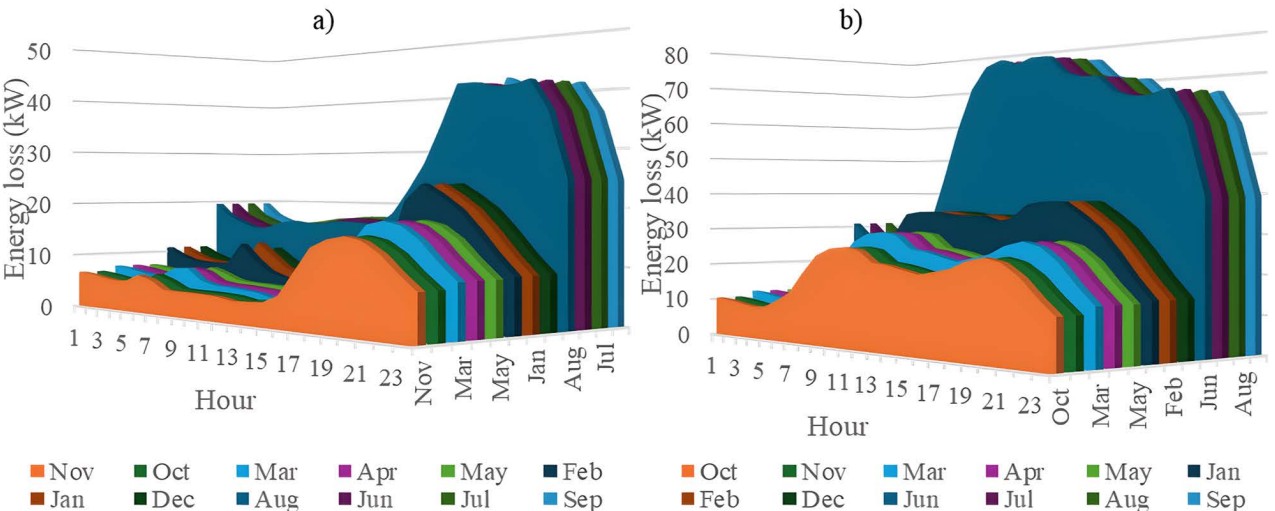

**Fig 28. The power loss value in an average day of the twelve months: a) Scenario 4, and b) the base system.**

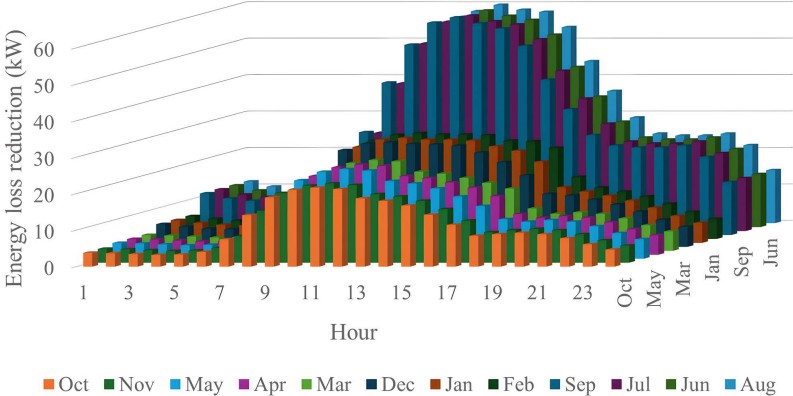

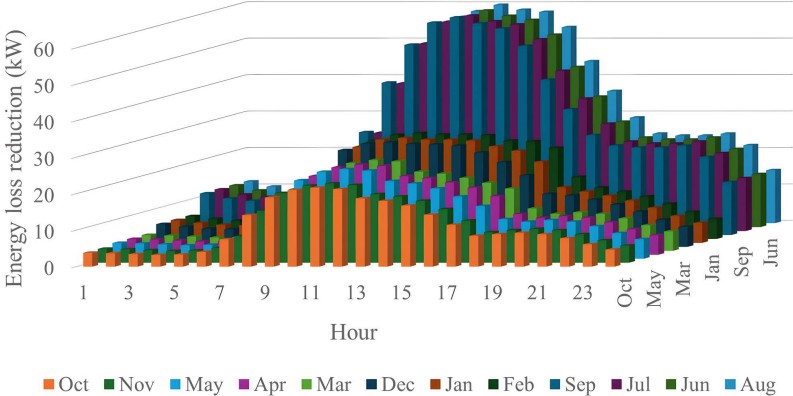

**Fig 29. The power loss reduction of each hour of the four months in the Scenario 4.**

With

$$Yi_y = Yi_0 \times (1 - DegF)^{y-1} \tag{36}$$

In the Equations (35) and (36), CAPEX is the capital expenditure cost in USD; $Pr_{O\&M}^{Fixed}$ is the fixed operation and maintenance price per kilowatt peak; $Pr_{O\&M}^{Varied}$ is the variable operation and maintenance price per kilowatt peak; $C_{InvRep}$ is the inverter replacement cost; $C_{Degr}$ is the decommissioning cost the ending year of the project's life time; $PR_{PVDG}$ is the rated power of the entire project; $Yi_y$ is the yielded energy of PVDGs in $y^{th}$ which is gradually decreased according Equation (36); $DegF$ is the degradation factor; and $r$ is the discount rate.

Note that the terms $\sum_{y=1}^{Y} \frac{Pr_{O\&M}^{Fixed} \times PR_{PVDG}}{(1+r)^y}$ and $\sum_{y=1}^{Y} \frac{Pr_{O\&M}^{Varied} \times Yi_y}{(1+r)^y}$ are respectively, total fixed operation and maintenance cost. CAPEX is achieved by by multiplying $Pr_{CAPEX}$ and $PR_{PVDG}$. Similarly, the $C_{InvRep}$ and $C_{Degr}$ are also determined by the price of each and the rated power of the project. The specific value of price terms and factors is shown in Table 9.

Table 10 provides a detailed calculation of all terms of cost that are needed to determine the EPC of the entire PVDGs project throughout 15 years of total life time, along with the total annual cost for each year. In the table, the total annual cost at the tenth year and fifth year is much higher than others due to these two years suffering from the inverter replacement cost and the decommissioning cost. Then the total annual cost will be compared to the case where the PVDG project is not implemented and shown in Table 11. As observed from the table, the total annual cost for the case without the PVDG project is actually the total electricity purchasing cost within a year determined by Equation (32), while the total annual cost for the case with the PVDG project is the electricity purchasing cost plus the entire project cost (EPC) as shown in Equation (33). Besides, the total accumulated is negative from the first year to the sixth year. In the eighth year, the accumulated profit begins to turn positive, and the last year of the entire project's lifetime yields a total profit of $ 8,480,854.37, which clearly demonstrates the economic feasibility of the entire project.

## 5. Conclusions

This study successfully employed two recently proposed meta-heuristic algorithms to optimize the allocation of DGs and SCBs for power loss reduction in different ACDN configurations, including the two standard systems, IEEE 69 and 85-node, and a 55-node practical ACDN in Vietnam. The placement of DGs and SCB is deployed in two phases: designing and operating. Particularly, in the designing phase, DGs and SCBs were optimized for placement on the three ACDN configurations using different settings for DGs as follows:

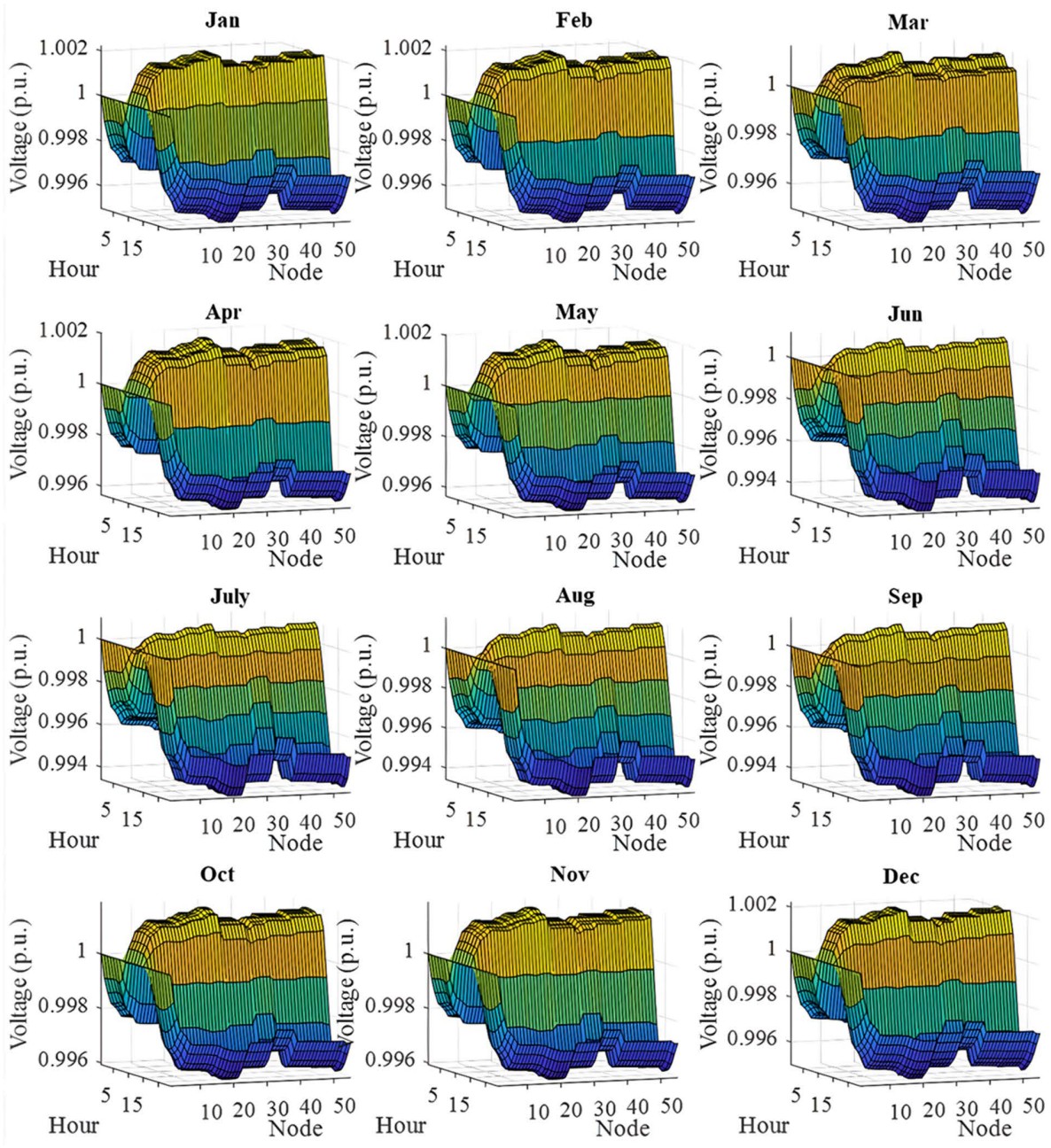

**Fig 30. The voltage profile within 24 hours in an average day of a) January, b) April, c) July, and d) October of Scenario 4.**

- For the standard ACDNs, two scenarios were tested: 1) DGs supplying only active power, and 2) DGs supplying both active and reactive power.

- Vietnam Practical ACDN, four scenarios were conducted: the first two scenarios are similar to the scenarios employed in standard ACDNs, while the remaining two explored different combined configurations of PVDGs and SCBs.

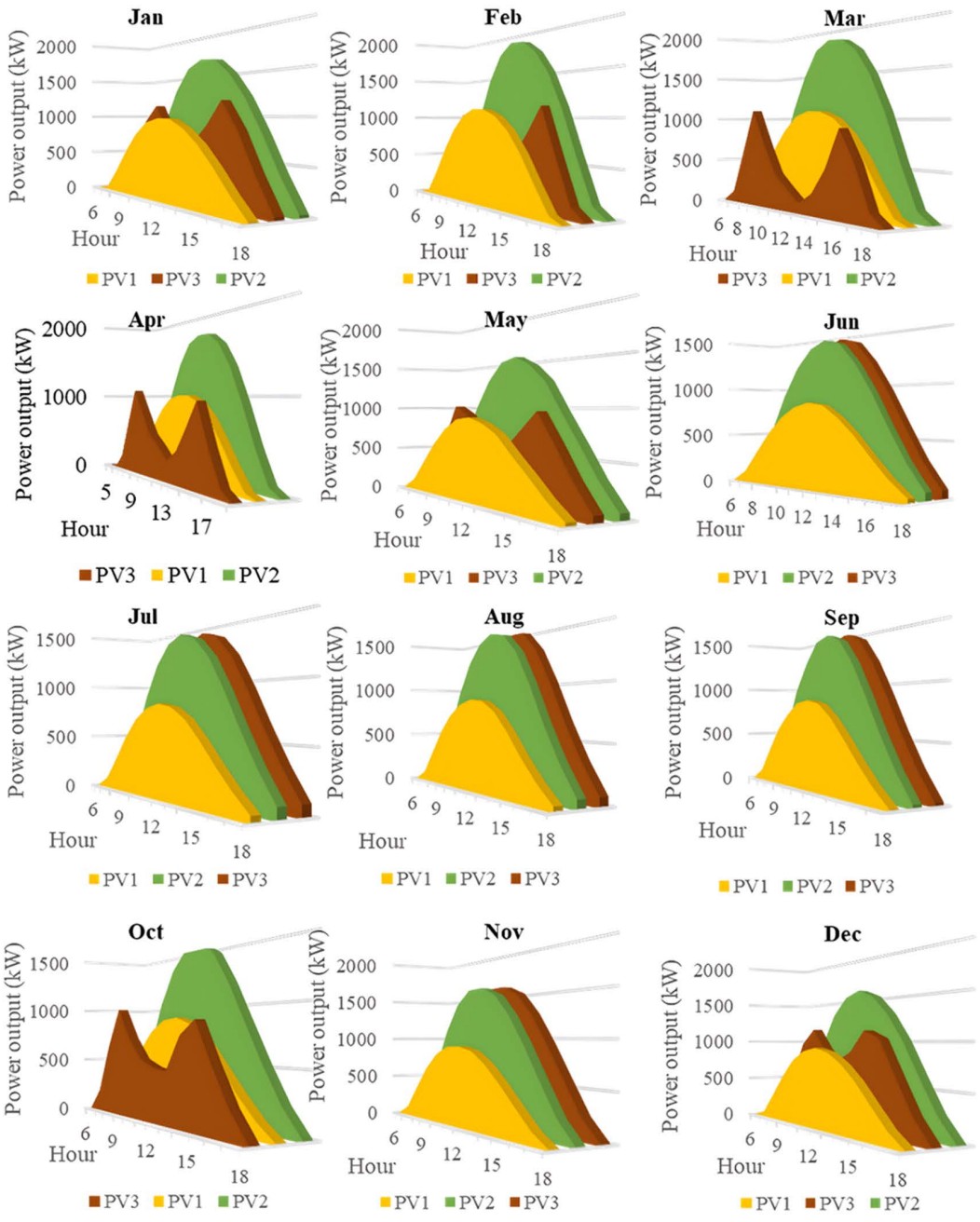

**Fig 31. The optimal power output of the three PVs while operating within 12 months.**

The employment of the designing phase aims to provide a particular merit about the effectiveness of placing DGs and SCBs in the three grids. Besides, the results achieved in the design phase are also the best foundation to justify the actual performance of the two applied algorithms. For instance, the results from this phase consistently demonstrated a significant reduction in power loss compared to the original grid. Crucially, the AOO has proven to completely outperform

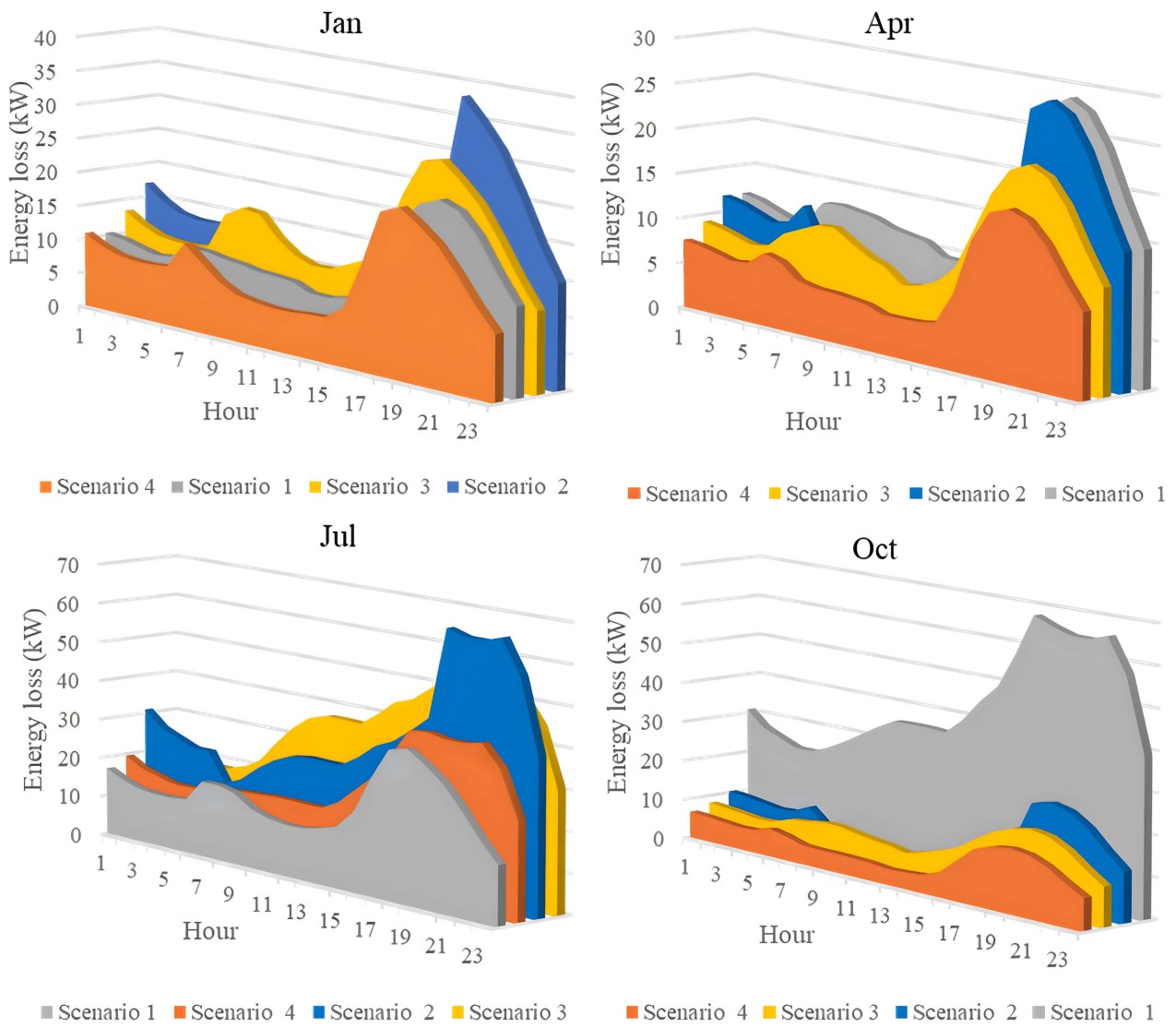

**Fig 32. The comparison on power loss values achieved in four scenarios of a) January, b) April, c) July, and d) October.**

**Table 8. Electricity price issued by TOUT.**

| Type | Time range | Voltage range [22, 110] kV |
|------|-----------|---------------------------|
| Off-peak price (USD) | 22:00–04:00 | 0.061 |
| Standard price (USD) | 04:00–09:30 | 0.11 |
| | 11:30–17:00 | |
| | 20:00–22:00 | |
| Peak price (USD) | 09:30–11:30 | 0.191 |
| | 17:00–20:00 | |

**Table 9. The specific values of the terms in Equations (35) and (36).**

| Terms | Value |
|---|---|
| $Y$ | 15 |
| $R$ [73] | 10 |
| $Pr_{CAPEX}$ ($/kWp) [74] | 810 |
| $Pr_{O\&M}^{Fixed}$ ($/kWp.year) [75] | 12 |
| $Pr_{O\&M}^{Varied}$ ($/kWh) [76] | 0.003 |
| Initial yield (kWh) | 10784300.1* |
| Degradation (%/year) [77] | 0.5 |
| Inverter replacement cost ($/kW) [78] | 20 |
| Degradation cost ($/kWp) [79] | 10 |
| Total rated power (kWp) | 7400** |

Note that * is the total yield energy of the three PVDGs in Scenario 4 taken from [69–71], while ** is determined by the total rated power of the three PVDGs in that scenario.

**Table 10. The presentation of all types of cost throughout 15 year of the considered PVDG project.**

| Year | CAPEX | Fixed O&M cost ($) | Variable O&M cost ($) | Inverter replacement cost ($) | Degradtion cost ($) | Electric purchasing cost ($) | Total Annual Cost ($) |
|---|---|---|---|---|---|---|---|
| 0 | 5994000 | 0 | 0.00 | 0 | 0 | 0.0 | 5994000.00 |
| 1 | 0 | 88800 | 32352.90 | 0 | 0 | 4595746.4 | 4716899.30 |
| 2 | 0 | 88800 | 32191.14 | 0 | 0 | 4595746.4 | 4716737.53 |
| 3 | 0 | 88800 | 32030.18 | 0 | 0 | 4595746.4 | 4716576.58 |
| 4 | 0 | 88800 | 31870.03 | 0 | 0 | 4595746.4 | 4716416.43 |
| 5 | 0 | 88800 | 31710.68 | 0 | 0 | 4595746.4 | 4716257.08 |
| 6 | 0 | 88800 | 31552.13 | 0 | 0 | 4595746.4 | 4716098.52 |
| 7 | 0 | 88800 | 31394.37 | 0 | 0 | 4595746.4 | 4715940.76 |
| 8 | 0 | 88800 | 31237.39 | 0 | 0 | 4595746.4 | 4715783.79 |
| 9 | 0 | 88800 | 31081.21 | 0 | 0 | 4595746.4 | 4715627.60 |
| 10 | 0 | 88800 | 30925.80 | 148000 | 0 | 4595746.4 | 4863472.20 |
| 11 | 0 | 88800 | 30771.17 | 0 | 0 | 4595746.4 | 4715317.57 |
| 12 | 0 | 88800 | 30617.32 | 0 | 0 | 4595746.4 | 4715163.71 |
| 13 | 0 | 88800 | 30464.23 | 0 | 0 | 4595746.4 | 4715010.62 |
| 14 | 0 | 88800 | 30311.91 | 0 | 0 | 4595746.4 | 4714858.30 |
| 15 | 0 | 88800 | 30160.35 | 0 | 74000 | 4595746.4 | 4788706.74 |
| Total Cost ($) | | | | | | | 76952866.73 |

the FGO across all tested scenarios, regardless of the expansion of the search space. The operating phase provides a detailed look at the results achieved in the designing phase, making it more reliable for a larger schedule.

Based on its superior performance, the AOO was selected to optimize the operational scheduling of PVDGs and SCBs, addressing the complex operational problem involving load demand variation and the dynamic power supply from PVDGs. The operating problem is solved on the 55-node Vietnam practical ACDNs with four scenarios, as mentioned in the design scenarios. Specifically, AOO was employed to determine the optimal hourly power outputs from PVDGs and SCBs for a

**Table 11. Total annual cost ($) of two cases with and without PVDGs project, benefit and accumulated profit.**

| Year | Total annual cost without PVDG (*) | Total annual cost with PVDG (**) | Benefit ($) (*) − (**) | Accumulated profit ($) |
|---|---|---|---|---|
| 0 | 0 | 5994000.00 | −5994000.00 | −5994000.00 |
| 1 | 5695581 | 4716899.30 | 978682.11 | −5015317.89 |
| 2 | 5695581 | 4716737.53 | 978843.87 | −4036474.02 |
| 3 | 5695581 | 4716576.58 | 979004.83 | −3057469.19 |
| 4 | 5695581 | 4716416.43 | 979164.98 | −2078304.20 |
| 5 | 5695581 | 4716257.08 | 979324.33 | −1098979.87 |
| 6 | 5695581 | 4716098.52 | 979482.88 | −119496.99 |
| 7 | 5695581 | 4715940.76 | 979640.65 | 860143.66 |
| 8 | 5695581 | 4715783.79 | 979797.62 | 1839941.27 |
| 9 | 5695581 | 4715627.60 | 979953.80 | 2819895.08 |
| 10 | 5695581 | 4863472.20 | 832109.21 | 3652004.29 |
| 11 | 5695581 | 4715317.57 | 980263.84 | 4632268.13 |
| 12 | 5695581 | 4715163.71 | 980417.69 | 5612685.82 |
| 13 | 5695581 | 4715010.62 | 980570.78 | 6593256.60 |
| 14 | 5695581 | 4714858.30 | 980723.10 | 7573979.70 |
| 15 | 5695581 | 4788706.74 | 906874.66 | 8480854.37 |
| Total profit ($) | | | 8480854.37 | − |

full day. The power supply data for PVDGs was derived from GSA with real solar radiation data representing an average day in the first month of the four quarters (January, April, July, and October). The operational results showed a significant reduction in energy loss, particularly during daytime when the PVDGs were active. While a reduction in energy loss was also observed during the night due to the support from SCBs, the isolated contribution of SCBs was found to be insignificant compared to the dramatic reduction achieved during the day when SCBs were combined with the active PVDGs. Additionally, the energy loss values through the four operational scenarios conducted on Vietnam practical ACDNs also indicate that operational Scenarios 4 offers the best energy loss overall during the three quarters of the year, except for the third quarter, with July being the first month. Specifically, operational Scenario 2 can offer a better hourly energy loss in the third quarter than the other three.

Besides the outstanding achievement and demonstration as clearly indicated across the results of the study, there are still several downsides that need to be seriously considered for better comprehensiveness in future works, as follows:

- The study considers placing DGs and SCBs across the whole grid except for node one, and does not take into account the availability of space in the field.

- The power loss reduction is achieved by placing DGs and SCBs only, while the other methods, such as network reconfiguration, placing distributed flexible alternative current transmissions (DFACTs), and battery energy storage systems (BESS), etc, are not employed.

- The study directly analyzes the available data provided by GSA to determine the actual power supplied by PVDGs at each hour, while the uncertainties of this renewable energy source are not considered.

- The combination of PVDGs and SCBs only proves its effectiveness in energy loss reduction during the day, while at night, the energy loss reduction is insignificant due to the deactivation of PVDGs.

## Supporting information

**S1 Table. The line data of the Chinfon feeder ACDN.** Detailed electrical parameters and configuration data of distribution lines used for the Chinfon feeder ACDN.
(XLSX)

**S2 Table. The load data of the Chinfon feeder ACDN.** Load demand data for buses in the Chinfon feeder ACDN used for simulation and analysis.
(XLSX)

## Author contributions

**Conceptualization:** Bon Nhan Nguyen.

**Data curation:** Minh Phuc Duong, Bon Nhan Nguyen.

**Formal analysis:** Valeriy Arkhincheev.

**Funding acquisition:** Bon Nhan Nguyen.

**Methodology:** Minh Phuc Duong.

**Software:** Minh Phuc Duong, Bon Nhan Nguyen.

**Validation:** Thang Trung Nguyen.

**Writing – original draft:** Minh Phuc Duong, Bon Nhan Nguyen.

**Writing – review & editing:** Valeriy Arkhincheev, Thang Trung Nguyen.

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
