## [Decision Letter · Decision Letter 0]

25 Nov 2025

Dear Dr. Nhan Bon,

Thank you for submitting your manuscript to PLOS ONE. After careful consideration, we feel that it has merit but does not fully meet PLOS ONE’s publication criteria as it currently stands. Therefore, we invite you to submit a revised version of the manuscript that addresses the points raised during the review process.

We look forward to receiving your revised manuscript.

Kind regards,

Zhengmao Li

Academic Editor

PLOS ONE

Journal Requirements:

“This work belongs to the project of the Year 2026 funded by Ho Chi Minh City University of Technology and Education, Vietnam. ”

“This work belongs to the project of the Year 2026 funded by Ho Chi Minh City University of Technology and Education, Vietnam.

The funder played a role in the payment of article publication charge, data collection, and report printing.”

“This work belongs to the project of the Year 2026 funded by Ho Chi Minh City University of Technology and Education, Vietnam.

The funder played a role in the payment of article publication charge, data collection, and report printing.”

d)        If you did not receive any funding for this study, please state: “The authors received no specific funding for this work.

5. We note that you have indicated that there are restrictions to data sharing for this study. PLOS only allows data to be available upon request if there are legal or ethical restrictions on sharing data publicly. For more information on unacceptable data access restrictions, please see http://journals.plos.org/plosone/s/data-availability#loc-unacceptable-data-access-restrictions.

Reviewers' comments:

Reviewer's Responses to Questions

**Comments to the Author**

1. Is the manuscript technically sound, and do the data support the conclusions?

Reviewer #1: Yes

Reviewer #2: Yes

2. Has the statistical analysis been performed appropriately and rigorously?

Reviewer #1: Yes

Reviewer #2: Yes

3. Have the authors made all data underlying the findings in their manuscript fully available?

Reviewer #1: No

Reviewer #2: Yes

4. Is the manuscript presented in an intelligible fashion and written in standard English?

Reviewer #1: Yes

Reviewer #2: Yes

Reviewer #1: Overall, the paper addresses a relevant problem and provides a comprehensive study of optimal placement and operation of PVDGs and SCBs using FGO and AOO on three distribution networks, including a 55-node feeder in Vietnam.

The two-phase structure (design versus operation) and the comparison of AOO against both FGO and a large set of existing algorithms give the work some depth, and the numerical results are clearly in favor of AOO.

To me, the narrative sometimes feels dense and a bit repetitive, and metaheuristics are not convincing enough. Please see my comments below:

1. In terms of logical flow, the Introduction is quite long and the transition from the broad literature review to the specific scope of “two-phase design and operation” is not always smooth. I have two suggestions here:

- For example, Section 1.3 lists “novelties” and then “main contributions” that overlap and could be merged into a more focused, shorter list, directly tied to what is actually done in Sections 2–4.

- You might also help readers by explicitly reminding them at the start of Section 4 how the design phase (peak-hour TAPL, equations (1)–(2)) and the operational phase (24-hour and quarterly energy-loss evaluation) fit together in the overall story.

2. Regarding the model design, why do you use a current-based formulation instead of power-flow-based expressions?

3. On the methodology side, the descriptions of FGO and AOO in Section 3 are mathematically detailed but a bit overwhelming. It might be very, very useful to give a short “high-level” explanation after each update rule, e.g., which parts are mainly responsible for global exploration versus local refinement.

4. Why do you use heuristic algorithms? In power system research, they always use MILP/LP to ensure the optimum of the results. You say your design is "optimal," but you are using a heuristic method, which is somewhat contradictory.

5. The choice of algorithmic parameters in Table 1 (e.g., N_pop = 20–40 and I_max = 200–300, with 50 test runs per case) is reasonable, but currently looks ad hoc. Can you justify? One suggestion (optional but nice to have): show convergence curves (best TAPL versus iteration) for at least one case, as well as average CPU times.

Reviewer #2: This study focuses on Vietnam's AC distribution network and uses two metaheuristic algorithms, FGO and AOO, to optimize the configuration of Photovoltaic Distributed Generators (PVDGs) and Shunt Capacitor Banks (SCBs) for reducing annual energy loss. Although it has a clear application scenario, there are multiple key flaws in research design, case logic, and data presentation, resulting in insufficient scientificity and rigor.

1. In the case of the 55-node power grid in Vietnam, the reactive power injection of SCBs in Case 4 (2070, 2250, 1230 kVar) is much higher than that in other cases, and the engineering feasibility of this configuration and its impact on power grid harmonics are not explained.

2. In the operational phase, only the average daily data of the first month of each quarter is selected, without considering the load fluctuation within the season and the photovoltaic output change under extreme weather, resulting in insufficient data representativeness.

3. When comparing AOO with other algorithms, only the numerical difference of TAPL is listed, and the comparative analysis of core performance indicators such as algorithm convergence speed and computational complexity is not provided.

4. In the case, the PVDG penetration rate is set to 25%, 50%, 75%, and 100%, but the basis for selecting this penetration rate range is not explained, and whether it is consistent with the local power grid access standards in Vietnam is not mentioned.

5. The constraint conditions do not consider the installation cost and maintenance cost of PVDGs and SCBs, and only take loss reduction as the goal, lacking economic feasibility analysis, which does not meet the actual engineering needs.

6. The original topological parameters of the 55-node power grid in Vietnam (such as line impedance and node load characteristics) are not fully presented, making it impossible to verify the authenticity and repeatability of the simulation results.

7. The discussion section does not explain why Case 4 performs worse than Case 2 in the third quarter (July), and fails to conduct in-depth analysis of the reasons combined with the seasonal changes of photovoltaic output and load characteristics.

**Do you want your identity to be public for this peer review?** For information about this choice, including consent withdrawal, please see our Privacy Policy

Reviewer #1: No

Reviewer #2: No

---

## [Author Response · Author response to Decision Letter 1]

13 Jan 2026

Dear Editor and Reviewers

The authors have added responses to the comments of the Editor Trisha Mae Tañedo Perez in the Cover letter. Please see the cover letter.

In the revised manuscript, we have clarified all comments and suggestions from Reviewers. All changes have been marked in blue and green in the revised manuscript. Responses to comments have been presented in the Responses to Reviewers file. Please find the file. We appreciate the efforts of the editor and reviewers in improving the manuscript. We hope the current form can satisfy your standards. Please see the Responses to Reviewer file.

Thank you very much.

Bon Nhan Nguyen

---

## [Decision Letter · Decision Letter 1]

22 Jan 2026

Optimal Design and Operation of Photovoltage Distributed Generators and Shunt Compensators for the Vietnam Alternative Current Distribution Network to Reduce Annual Energy Loss

PONE-D-25-59933R1

Dear Dr. Nhan Bon,

We’re pleased to inform you that your manuscript has been judged scientifically suitable for publication and will be formally accepted for publication once it meets all outstanding technical requirements.

Kind regards,

Zhengmao Li

Academic Editor

PLOS One

Additional Editor Comments (optional):

Reviewers' comments:

Reviewer's Responses to Questions

**Comments to the Author**

Reviewer #1: All comments have been addressed

Reviewer #2: All comments have been addressed

2. Is the manuscript technically sound, and do the data support the conclusions?

Reviewer #1: Yes

Reviewer #2: Yes

3. Has the statistical analysis been performed appropriately and rigorously?

Reviewer #1: Yes

Reviewer #2: Yes

4. Have the authors made all data underlying the findings in their manuscript fully available?

Reviewer #1: (No Response)

Reviewer #2: Yes

5. Is the manuscript presented in an intelligible fashion and written in standard English?

Reviewer #1: Yes

Reviewer #2: Yes

Reviewer #1: (No Response)

Reviewer #2: The author responded well to my questions, the article can be accepted, I hope the author 's future work can be better and better, congratulations

**Do you want your identity to be public for this peer review?** For information about this choice, including consent withdrawal, please see our Privacy Policy

Reviewer #1: No

Reviewer #2: No

---

## [Editor Report · Acceptance letter]

PONE-D-25-59933R1

PLOS One

Dear Dr. Nguyen,

I'm pleased to inform you that your manuscript has been deemed suitable for publication in PLOS One. Congratulations! Your manuscript is now being handed over to our production team.

Kind regards,

on behalf of

Dr Zhengmao Li

Academic Editor

PLOS One